# Structural insights into p300 regulation and acetylation-dependent genome organisation

Ziad Ibrahim[1,5], Tao Wang[2], Olivier Destaing [2], Nicola Salvi [3], Naghmeh Hoghoughi[2], Clovis Chabert[2], Alexandra Rusu[1], Jinjun Gao[4], Leonardo Feletto[1], Nicolas Reynoird [2], Thomas Schalch[1], Yingming Zhao[4], Martin Blackledge [3], Saadi Khochbin [2] & Daniel Panne [1] ✉

Histone modifications are deposited by chromatin modifying enzymes and read out by proteins that recognize the modified state. BRD4-NUT is an oncogenic fusion protein of the acetyl lysine reader BRD4 that binds to the acetylase p300 and enables formation of long-range intra- and inter-chromosomal interactions. We here examine how acetylation reading and writing enable formation of such interactions. We show that NUT contains an acidic transcriptional activation domain that binds to the TAZ2 domain of p300. We use NMR to investigate the structure of the complex and found that the TAZ2 domain has an autoinhibitory role for p300. NUT-TAZ2 interaction or mutations found in cancer that interfere with autoinhibition by TAZ2 allosterically activate p300. p300 activation results in a self-organizing, acetylation-dependent feed-forward reaction that enables long-range interactions by bromodomain multivalent acetyl-lysine binding. We discuss the implications for chromatin organisation, gene regulation and dysregulation in disease.

Chromosome capture experiments have led to the identification of two major mechanisms that are behind large-scale chromosome organisation[1,2]: (1) at shorter genomic distances, genomes are organized into topologically associating domains[3,4] which arise through the process of loop extrusion by cohesin; (2) at longer-range genomic distances, genomes are spatially compartmentalised into active, euchromatic and inactive, heterochromatic regions[5,6].

One of the main mechanisms driving genome compartmentalisation is thought to be selective self-association of active or inactive regions and physical constraints including association with architectural elements[7,8]. One hypothesis is that 'readers' of histone modifications seed 'condensation' reactions as a function of the histone modification state[6,9]. It has been proposed that compartmentalisation arises specifically due to attractions between heterochromatic regions, possibly driven by polymer-polymer interactions and liquid-liquid phase separation (LLPS)[10,11]. Whether heterochromatic compartments form through LLPS or other mechanisms is controversial[12-15].

Histone acetylation counteracts heterochromatinization and has long been associated with chromatin decondensation and euchromatic chromatin regions[16]. Targeting of heterochromatin with acidic transcription factor (TF) activation domains (ADs) or treatment with deacetylase inhibitors results in hyperacetylation, decondensation and relocation to the nuclear interior even in the absence of active transcription[17-23].

Numerous TFs associate through their ADs with a small set of co-activators including Mediator and the acetylases CBP and its paralogue p300 (also known as KAT3A and KAT3B respectively; hereafter 'p300' for short)[24-29]. p300 has long been known to be important in integrating multiple signal transduction pathways in the nucleus and enabling enhancer-mediated transcription[30-32]. Intriguingly, it has been proposed

[1]Leicester Institute of Structural and Chemical Biology, Department of Molecular and Cell Biology, University of Leicester, Leicester, UK. [2]CNRS UMR 5309, INSERM U1209, Université Grenoble Alpes, Institute for Advanced Biosciences, Grenoble, France. [3]Institut de Biologie Structurale, CNRS, CEA, UGA, Grenoble, France. [4]Ben May Department of Cancer Research, The University of Chicago, Chicago, IL 60637, USA. [5]Present address: Department of Structural Biology, St. Jude Children's Research Hospital, Memphis, United States. ✉e-mail: daniel.panne@leicester.ac.uk

that spatial sequestration of heterochromatin at the nuclear lamina requires TF-mediated retention of p300 in euchromatin[33]. Quantitative Mass Spectrometry experiments show that p300 is present at limiting concentrations in the nucleus while HDAC co-repressors are in large excess[34]. As p300 is responsible for up to a third of nuclear acetylation[35], TF-mediated competition for and retention of these essential and limiting co-activators in euchromatic regions could be a key driver event that not only shapes gene specific transcription but also eu- and heterochromatic genome compartmentalisation.

It has been proposed that transcriptional control may be driven by the formation of phase-separated condensates[36]. Accordingly, condensates of TF ADs[37], co-activators and the transcriptional machinery have all been proposed to be relevant for transcriptional regulation[38–43]. p300 contains an ordered catalytic 'core' domain including a HAT- and bromodomain (BD) and several domains that form promiscuous interactions with the disordered ADs of hundreds of cellular TFs (Fig. 1a)[44,45]. These interactions domains are separated by long, intrinsically disordered regions (IDRs). p300 is activated by autoacetylation[46], a reaction that can be controlled by cellular signalling through TF activation and dimerisation[47]. The p300 core contains a RING and TAZ2 domain that have an autoinhibitory function[44,47]. Perturbation of the autoinhibitory function of the RING and TAZ2 domains by mutation leads to constitutive p300 activation and formation of nuclear droplets that co-localise with p53. These droplets are dispersed upon HAT or BD inhibition, suggesting that a form of phase separation (we here use the more general term 'condensation') occurs through an acetylation-dependent 'read-write' mechanism[47]. Other data indicate that p300 forms nuclear condensates through the 'core' domain[48]. Co-condensation of p300 with TFs on enhancer loci is thought to enable activation of p300's catalytic activity thus allowing the recruitment of acetylation-dependent coactivators, including the BD−and extraterminal domain-containing (BET) protein BRD4[49].

BET proteins have long been associated with the control of large-scale nuclear structure and chromatin organization. BRDT, a testes-specific BET, mediates histone acetylation-dependent chromatin condensation[50] and is required for stage-specific gene expression and acetylation-dependent histone removal during spermatogenesis[51–53]. Similarly, BRD4 has been linked both to gene-specific[54] and higher-order chromatin organisation[55,56]. In yeast, the BET protein BDF1 maintains physical genome barriers by preventing spreading of heterochromatin[57]. In vitro data show that under specific buffer conditions, chromatin can undergo LLPS which can be modulated by acetylation and BRD4 binding[58].

BRD4 also forms condensates in cells[39]. In vitro and in vivo experiments with systems inducing high local concentration suggest that a C-terminal IDR drives the condensation reaction[39,59]. However, a BRD4 isoform lacking the C-terminal IDR also forms condensates in cells[60]. Other data indicate that BRD4 clustering does not depend on IDR-mediated condensation but on DNA and BD-acetyl-lysine recognition which possibly enables long-range enhancer-promoter communication[61]. At a larger scale, such acetylation-dependent long-range interactions could enable selective self-association of euchromatic regions and provide a link between genome topology and gene regulation. It is therefore important to understand the underlying 'read-write' mechanisms that drive such long-range interactions.

BRD4-NUT is an oncogenic fusion protein that drives NUT carcinoma, one of the most aggressive human solid malignancies known[62]. NUT is normally expressed in post-meiotic spermatogenic cells, where it interacts with p300 and triggers genome-wide histone hyperacetylation, which enables recruitment of BRDT, histone-to-protamine exchange and chromatin compaction during spermatogenesis[53]. In the oncogenic fusion protein, NUT binds to and activates p300 which drives an acetylation- and BRD4 BD-dependent 'feed-forward' loop that leads to formation of hyperacetylated nuclear 'foci' visible by light microscopy[62–64], that resemble foci seen in cells expressing constitutively activated p300[47]. These foci correspond to large chromosomal 'megadomains' (100 kb to 2 Mb) that are enriched for BRD4-NUT, p300 and acetylated histones[65–67]. Megadomains form in the absence of transcription and engage in very long-range acetylation-dependent inter- and intrachromosomal interactions[68].

The BRD4-NUT model thus exemplifies how a modification of BRD4 enables recruitment and activation of p300 and establishment of a self-organizing feed-forward reaction that results in chromatin hyperacetylation and long-range contact. The underlying mechanisms could have broad implications for our understanding of how 'reading' and 'writing' of chromatin acetylation control gene expression and how dysregulation of this process leads to aggressive forms of disease.

Here we use the BDR4-NUT/p300 system to dissect the mechanism that drives large-scale chromatin organization. We determine the element of BRD4-NUT required for interaction with p300. We find that the NUT portion of the fusion protein contains an acidic AD that interacts with the p300 TAZ2 domain, a common binding site for numerous TF ADs. We investigate the solution behaviour of the complex and find that interaction with the TAZ2 domain stimulates p300 HAT activity. Our data indicate that the TAZ2 domain has an autoinhibitory function that is relieved upon TAZ2 ligand-binding. We systematically probe which elements of BRD4-NUT and p300 drive large scale chromatin condensation in cells. We find that condensation depends on a feed-forward mechanism that involves allosteric activation of p300 by binding of NUT to the p300 TAZ2 domain, nucleation of the condensation reaction through BRD4 and p300 BD-acetyl-lysine binding and that BD multivalency is required. We find that Adenoviral E1A, another TAZ2 ligand, or mutations associated with cancer that interfere with TAZ2 autoinhibition also activate p300. Our data reveal how activation of p300 establishes a self-organising positive feedback mechanism and how an acetylation-dependent read-write mechanism drives larger-scale chromatin architecture.

## Results

### Identification of an acidic activation domain in NUT

Previous data showed that a region spanning BRD4-NUT amino acids 347-588 (NUT F1c; NUT numbering is used; 1060-1307 in BRD4-NUT) directly interacts with p300 (Fig. 1a, b)[63]. To map the interaction further, we produced a series of deletions within NUT and assayed their interaction with p300 using co-sedimentation and GST pulldown assays. NUT fragments that contained amino acids 396-482 generally co-sedimented with the p300 core-CH3, the isolated p300 CH3 or TAZ2 domains (Supplementary Figs. 1a, b; 2a–f).

The TAZ2 domain forms dynamic interactions with short ~30 amino acid transcription factor activation domains (ADs) that are unrelated in sequence but generally share net negative charge, hydrophobic, and aromatic residues that are typically intrinsically disordered (IDR) in isolation[45]. Accordingly, sequence analysis revealed that the region of NUT is predicted to be disordered, and negatively charged with modest propensity to self-aggregate (Fig. 1c). A luciferase assay confirmed that the isolated NUT region can stimulate transcription (Supplementary Fig. 1c), in agreement with previous data showing that tethering of NUT on chromatin is sufficient to induce transcriptional activation[67]. We therefore conclude that NUT contains an acidic AD that directly interacts with the TAZ2 domain of p300 (Fig. 1b).

The high percentage of negatively charged amino acid residues in ADs is thought to prevent hydrophobic collapse while the distribution of charges maintains the conformational ensemble in a more expanded state leaving the charged and hydrophobic side chains exposed to the solvent and binding partners[69]. Accordingly, analysis by size-exclusion chromatography−multi-angle laser-light scattering (SEC−MALS) showed that a NUT fragment spanning amino acids 347-480 (NUT4) is monomeric and monodisperse in solution with a large hydrodynamic radius, consistent with an extended conformation

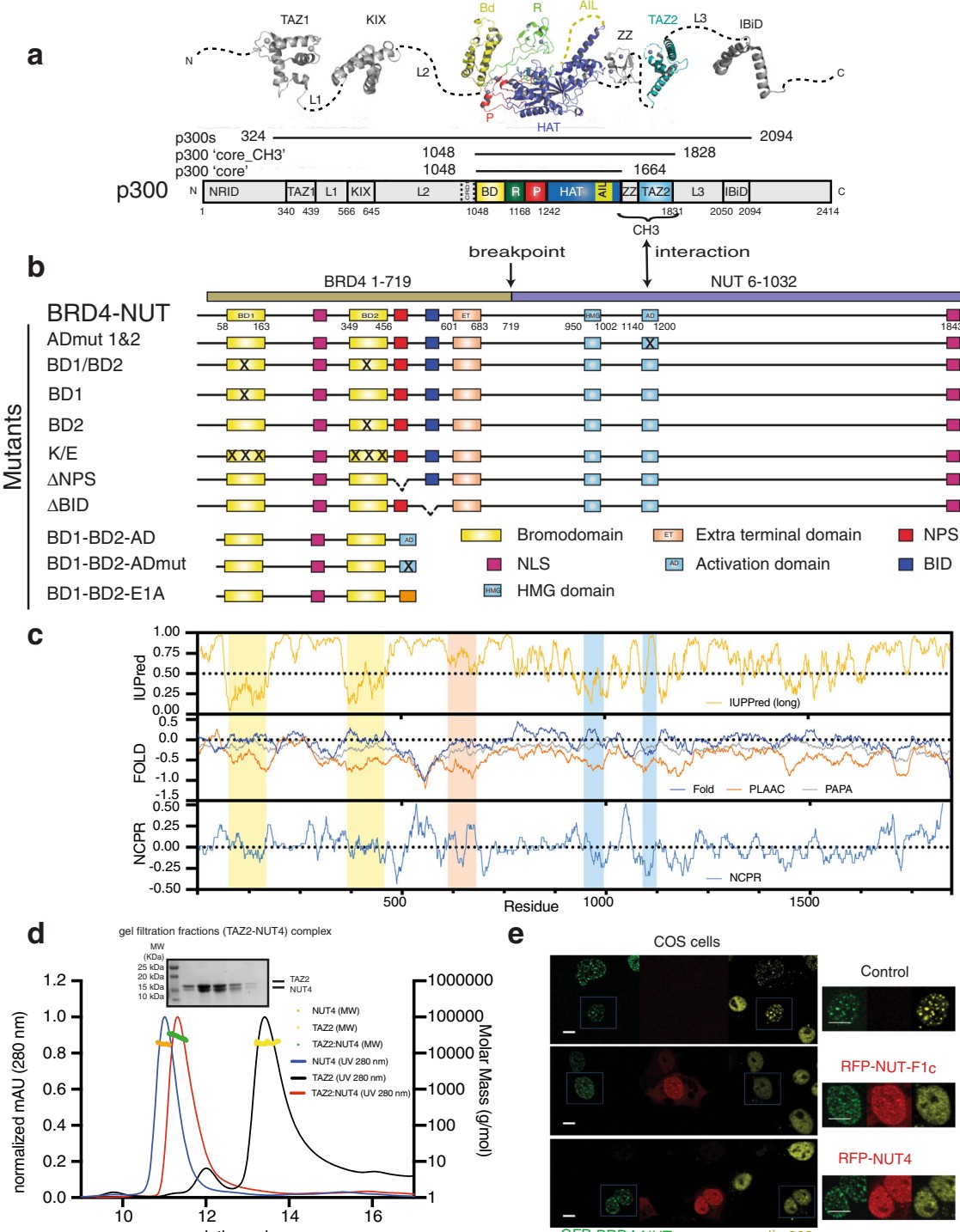

**Fig. 1 | Identification of an acidic activation domain in NUT. a** Domain structure of p300 and **b** BRD4-NUT constructs. **c** IUPred, prediction of intrinsic disorder; FOLD, folding prediction using the foldindex (blue), PLAAC (Prion-Like Amino Acid Composition; orange) and PAPA (Prion Aggregation Prediction Algorithm; grey) analysis, NCPR (Net Charge Per Residue) with a sliding window of ten residues of the BRD4-NUT protein. **d** SEC-MALLS analysis of the TAZ2 domain of p300 alone (black), NUT4 alone (blue) and the complex between TAZ2 and NUT4 (red). SDS-PAGE analysis of the peak fractions of the TAZ2-NUT4 complex is shown. The experiment was repeated twice with consistency. **e** Competition analysis: Cos7 cells were transfected with constructs expressing GFP-BRD4-NUT (green) alone or co-transfected with RFP-NUT-F1c and RFP-NUT4 (red). Immunofluorescence with anti-p300 antibody was used to visualize endogenous p300. Scale bars, 10 μm. The experiment was repeated three times with consistency.

(Fig. 1d). The TAZ2:NUT4 complex formed a 1:1 complex with a reduced hydrodynamic radius, indicative of conformational compaction upon binding (Fig. 1d).

Co-expression of NUT F1c can prevent condensate formation in the nucleus by acting as a dominant negative inhibitor for the p300-NUT interaction[63,67]. Accordingly, overexpression of RFP-NUT F1c, -NUT4, or -NUT2 but not -NUT1 or -NUT3 reduced p300 foci in COS cells (Fig. 1e, Supplementary Fig. 2g). As AD function is usually confined to short sequences, we further mapped the interaction by limited proteolysis followed by in-gel acid hydrolysis and mass spectrometry

(Supplementary Fig. 1d). Fragments spanning amino acids 396-482 (NUT9) or a peptide spanning 458-482 (NUT10) all bound to the TAZ2 domain (Supplementary Table 1).

## TAZ2-NUT interaction

NMR spectra of NUT7, spanning 448-492, feature little dispersion of $^{1}$H resonances (Fig. 2a), indicating that the protein is largely disordered in solution. We used a series of 3D experiments (see Methods) to assign most of the backbone resonances of NUT7 (Supplementary Fig. 3a–c). Secondary structure propensities (SSP) calculated using $^{13}$Cα, $^{13}$C′ and $^{15}$N chemical shifts show that amino acids 451-472 have a significant tendency to sample helical conformations in solution (Fig. 2b). These NMR studies of the NUT7 AD allow us to conclude that this domain is an IDR in isolation with a pre-organised N-terminal α helix.

We monitored binding to the TAZ2 domain by collecting a series of $^{1}$H, $^{15}$N-HSQC spectra of $^{15}$N-labelled NUT7 mixed with increasing concentrations of TAZ2 (Fig. 2a). Chemical shift perturbations (CSP) showed that several NUT7 backbone amide resonances, including A450, L451, G460-A464, V467 and E475 were perturbed upon addition of an equimolar amount of TAZ2 while resonances from the C-terminus of NUT7 do not appear to shift significantly upon addition of TAZ2 (Fig. 2c, Supplementary Table 2). Thus, only the N-terminal α-helical segment of NUT7 is involved in TAZ2 binding. Addition of equimolar amounts of unlabelled NUT7 to $^{15}$N-labelled TAZ2 showed significant CSPs of resonances corresponding to residues R1737, I1739, M1761, R1763, V1765, Q1766, T1768, K1783, Q1784, I1809 (Fig. 2a, c), that are mostly located on the surface of TAZ2. These site-specific measurements reveal that the region from 450-470, exhibiting α-helical propensity in the free form, binds to the surface of TAZ2.

To understand the binding mode in more detail, we measured both $^{13}$C- and $^{15}$N-HSQC-NOESY spectra on a sample of $^{15}$N,$^{13}$C-labelled TAZ2 to which one equivalent of unlabelled NUT7 was added. We also measured $^{15}$N-HSQC-NOESY spectra on $^{15}$N-labelled NUT7 in admixture with one equivalent of unlabelled TAZ2. The NOESY spectra contain very few cross-peaks that can be unambiguously assigned to inter-molecular contacts most likely due to extensive dynamic fluctuations of NUT7 even in the bound state. One contact is nevertheless detected between the backbone amide hydrogen of S1726 in TAZ2 and the backbone amide hydrogen of residue L463 (3.9 +/− 0.8 Å) and the Hβ of A464 (3.5 +/− 0.7 Å) in NUT7. Because of relative sparsity and ambiguity of structural restraints, we used HADDOCK[70] to derive a structural model of the interaction consistent with all of the observed spectroscopic changes, combining chemical shifts and sparse NOEs (Fig. 2d, e). NMR restraints and structural statistics are summarized in Supplementary Table 3. On the basis of this model, it appears that the N-terminus of NUT7 folds into an amphipathic helix that binds a hydrophobic groove in the interface between helices α1/α2/α3 on the surface of TAZ2 (Fig. 2d). Only the NUT backbone residues P448 to A464 are well ordered while residues at the C-terminus (Q465-D492) do not interact with TAZ2 and remain disordered. A set of hydrophobic side chains of NUT L449, A450, L451, L455, L461 are in contact with TAZ2 residues I1735, M1761, V1764, I1781 (Fig. 2d). The interaction is further stabilised by electrostatic interactions involving NUT residues E453, E454, E456, E458, E459 and TAZ2 residues R1737, S1754, S1757 and K1760 (Fig. 2d). In summary, similar to other TAZ2:AD interactions, a positively charged TAZ2 domain interacts with a negatively charged NUT AD (Fig. 2e). The key interfacial binding surfaces are conserved in p300 and NUT (Fig. 2e).

## Analysis of the TAZ2:NUT interaction

As ADs are frequently bi-partite and the NUT7 fragment in our structure only partially engages the TAZ2 binding surface, we further explored the region N-terminal of NUT7. We found using HSQC competition, that the NUT7 fragment is unable to displace p53 from the surface of TAZ2 when added to the complex (Supplementary

Fig. 4e). Biolayer Interferometry (BLI) experiments showed that NUT8 (396-470) bound to TAZ2 with a dissociation constant of $K_d = 0.96 +/− 0.29\ \mu$M (Fig. 3a, b). Alanine substitutions of conserved residues in the region spanning 447-462 (mut3, Fig. 3c), the main binding epitope in NUT7, did not abolish TAZ2 binding in GST-pull-downs, but reduced TAZ2 binding affinity (Fig. 3b, Supplementary Fig. 4a, b). Additional Alanine substitution of L436, L440, L441 (mut4, Fig. 3c) further reduced TAZ2 affinity (Fig. 3b). In contrast, binding was largely abolished by additional mutation to Alanine of L410, L411, Y413, L427, C428, F423, V424, V427, V430 and I431 (mut1, Fig. 3b–d, Supplementary Fig. 4a). NUT mut1 was a bit more instable as compared to the equivalent WT fragment after purification (see Source Data, Fig. 4c). As alanine repeat expansion can alter transcription factor condensation properties[71], we also generated a NUT variant containing mostly Glycine, Serine and charge reversal mutations (mut2, Fig. 3c). Mut2 also failed to bind TAZ2 in BLI and GST-pulldown experiments (Fig. 3b–d, Supplementary Fig. 4a).

As mutation of isolated binding epitopes reduced but did not abolish TAZ2 binding, we conclude that the NUT AD is made up of subdomains that contain short amphipathic interaction motifs that synergise to enable high affinity binding. TF ADs frequently bind p300 through a minimal ΦΧΧΦΦ motif (X is any residue and Φ is hydrophobic), frequently LXXLL[72]. The NUT AD contains several such conserved motifs most of which are predicted to engage TAZ2 (Fig. 3c, e).

In our NMR-based model, the isolated AD sub-domain in NUT7 is positioned at the interface of the TAZ2 α1, α2 and α3 helices, which frequently functions as a docking site for TF ADs that typically adopt helical structure upon binding (Supplementary Fig. 4f). AlphaFold (AF) computational models predict 5 helices for NUT (Fig. 3e) that engage the protein interaction surfaces of TAZ2 (Fig. 3f, Supplementary Fig. 4c, g)[73,74]. In the model, the NUT7 AD is re-positioned while the TAZ2 α1, α2, α3 surface is occupied by α3 of the NUT AD. Structure determination of the TAZ2 domain in complex with the entire NUT AD was not possible as NMR spectra suffered from severe broadening. As (1) isolated AD sub-domains can interact with alternate TAZ2 surfaces[75] and (2) as the residues that impair binding are located in the predicted TAZ2-NUT binding interface, these data provide a reasonable model for NUT-p300 interaction.

## The role of the TAZ2 domain in p300 regulation and substrate acetylation

Previous data show that NUT stimulates p300 auto- and histone acetylation in cells and in vitro[53,63]. The AF model of p300 suggests that the TAZ2 domain is positioned over the p300 HAT active site (Fig. 4a). The TAZ2 C-terminal helix (amino acids 1806-1836) is sandwiched between the RING and HAT domains proximal to the lysine substrate-binding tunnel in a conformation that is incompatible with substrate access. In contrast, a model in the presence of NUT AD shows that the TAZ2 domain is displaced from the HAT active site (Fig. 4b). These computational models suggest that the TAZ2 domain auto-inhibits the HAT domain and that this auto-inhibition is relieved upon ligand-binding to the TAZ2 domain. As the TAZ2 domain binds numerous transcription factor ADs, such a TAZ2-mediated control mechanism may be broadly relevant and a key aspect of how TFs control chromatin acetylation and gene regulation.

To test this model, we incubated increasing amounts of GST-NUT8wt with p300s (amino acids 324-2094) in the presence of [$^{14}$C] acetyl coenzyme A. Addition of NUT WT but not mut1 stimulated p300 autoacetylation in a dose-dependent manner (Fig. 4c). Despite the identical number of lysine amino acids, only NUT WT but not mut1 was acetylated by p300. Other NUT fragments that contain the AD also activated p300 (Supplementary Fig. 5a, b). In agreement with the computational model that AD-TAZ2 interaction relieves autoinhibition, acetylation reactions with the core domain of p300, lacking TAZ2 (Fig. 1a; p300 amino acids 1048-1664), showed that p300

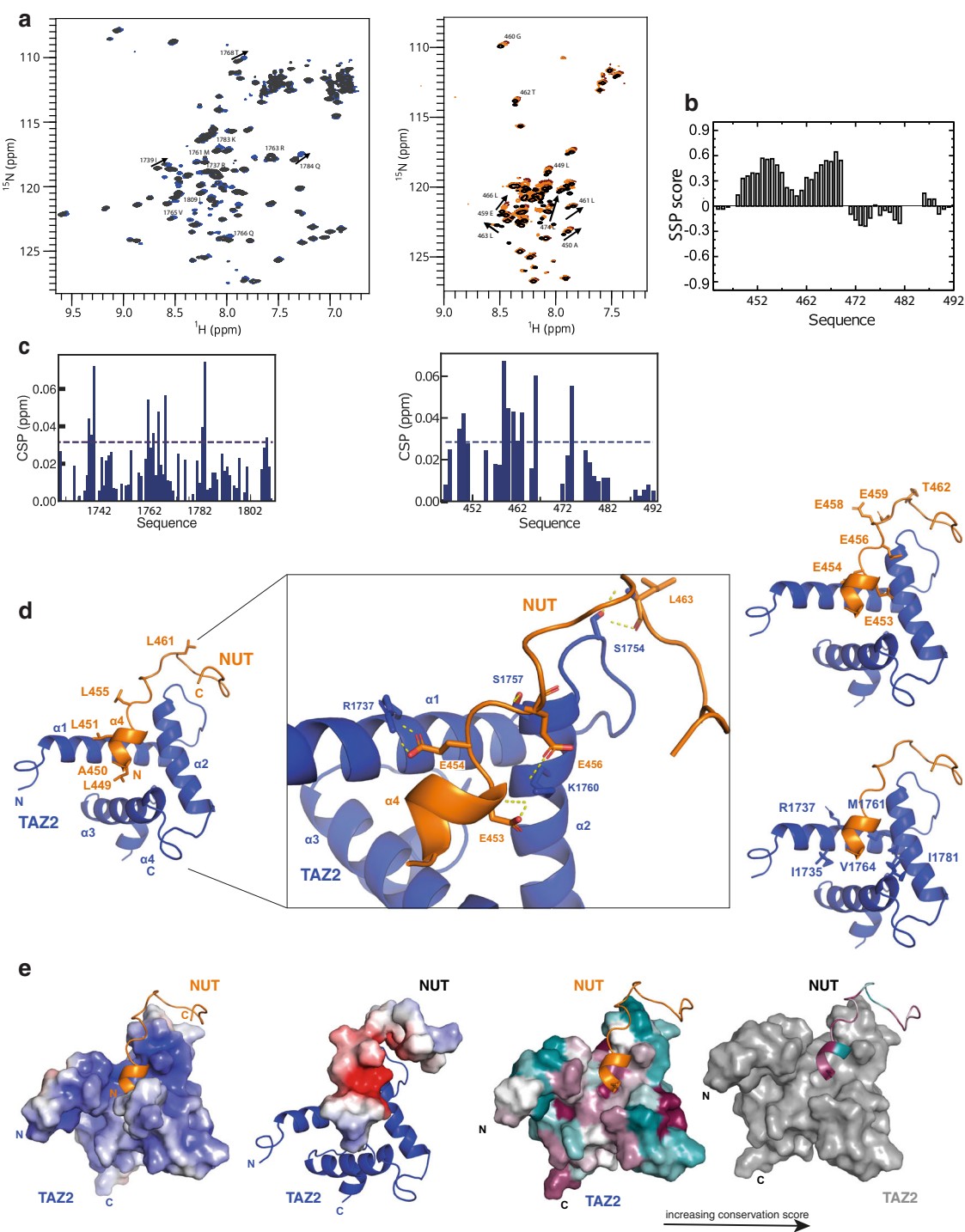

**Fig. 2 | TAZ2:NUT interaction. a** (Left) Overlay of ¹H-¹⁵N HSQC spectra of free TAZ2 (black) and after addition of one molar equivalent of NUT7 (448-492) (blue). Arrows indicate chemical shift changes for individual cross-peaks. (Right) Overlay of ¹H-¹⁵N HSQC spectra of free NUT7 (448-492) and after adding half (orange) and one (brown) molar equivalent of TAZ2. Arrows indicate chemical shift changes for individual cross-peaks. **b** Secondary structure propensities (SSP) plot of the NUT (448-492) fragment in its free state. positive values on the Y-axis indicate a-helical propensity, the X-axis is the amino acid sequence. **c** (Left) Chemical shift perturbations on TAZ2 upon adding one molar equivalent of unlabelled NUT7 (448-492). (Right) Chemical shift perturbations on NUT7 (448-492) upon adding one molar equivalent of unlabelled TAZ2. **d** (Left) Ribbon representation of lowest energy NMR model of the TAZ2 (blue)-NUT (orange) complex. Hydrophobic residues on the surface of NUT are shown. (middle) Zoom-in view of the interface between NUT (orange) and TAZ2 (blue). Residues on both sides implicated in the interaction as well as the hydrogen bonding network are shown. (Right) acidic residues on the surface of NUT (top) as well as hydrophobic and basic residues on the surface of TAZ2 (bottom) are shown. **e** TAZ2 and NUT7 (448-492) are surface rendered and coloured according to electrostatic potential (left) and sequence conservation (right).

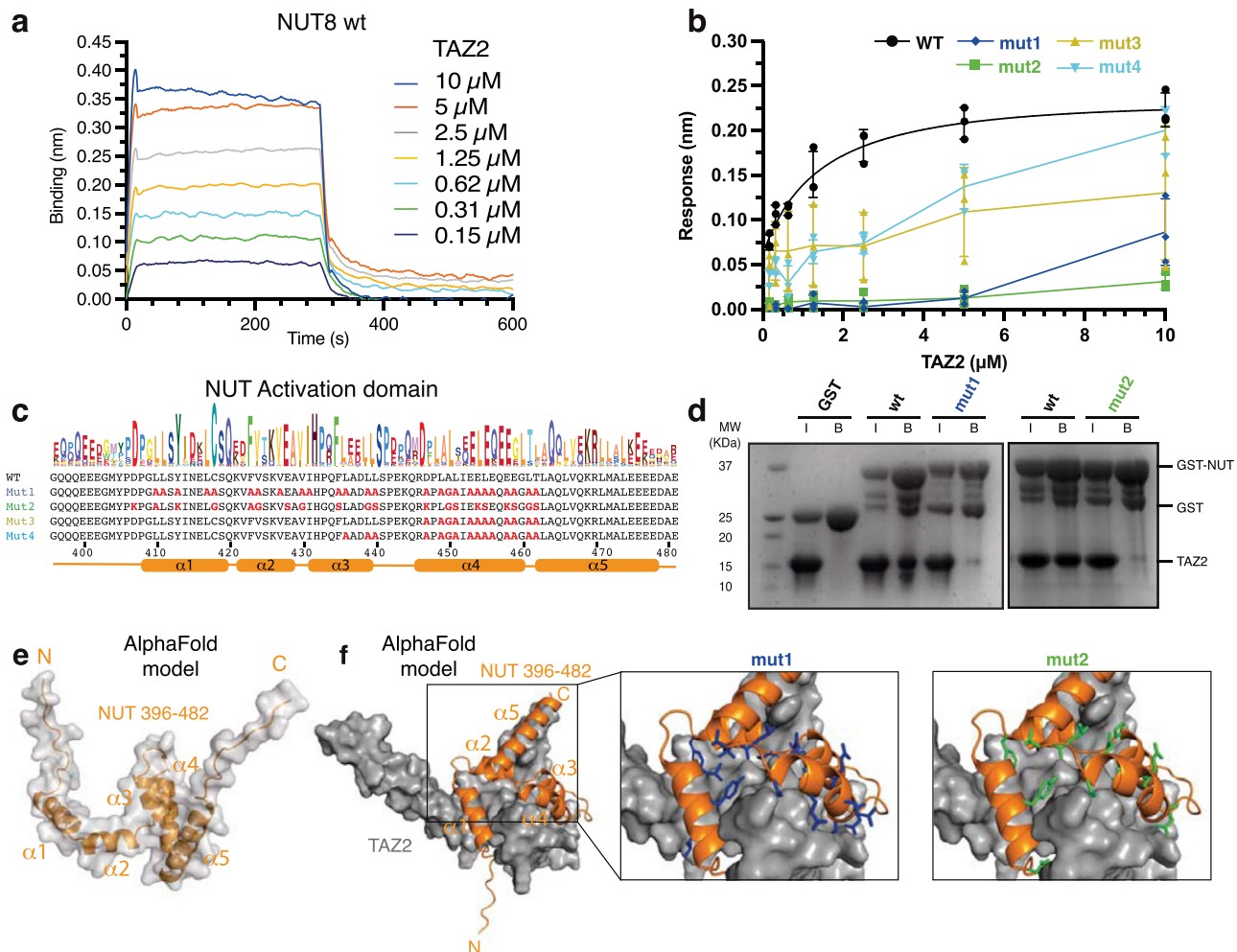

**Fig. 3 | Analysis of the TAZ2-NUT interaction. a** Representative Biolayer Interferometry (BLI) profiles of p300 TAZ2-NUT binding from which the steady-state data were derived. Colours indicate different concentrations of TAZ2 used. **b** Steady state analysis of p300 TAZ2 interaction with wild-type (black) and NUT mutants using BLI. Data are presented as mean values +/− SD. $N = 3$ independent technical replicate experiments were done. **c** Sequences of the NUT AD and mutants (in bold red) used. Top: Skylign plot showing NUT AD sequence conservation; Bottom: predicted secondary structure. **d** GST pulldown analysis of TAZ2 and NUT8 (396-470) wild type and mutants 1 and 2. The experiment was repeated three times with consistency. **e** AlphaFold prediction of NUT (396-482) structure. **f** Left: AlphaFold prediction of the TAZ2-NUT complex structure. Right: mutated residues in NUT mutant 1 (blue) and 2 (green) are shown.

autoacetylation is not stimulated by the presence of NUT WT and both WT and mutant substrates are acetylated at equal rates (Fig. 4d). Another classic TAZ2 ligand, Adenoviral E1A also showed a dose-depended stimulation of p300s auto- and substrate acetylation (Fig. 4e). Together, domain truncation and mutagenesis support the computational models but further validation is required.

Recent data indicate that p300 directly engages nucleosome core particles (NCPs) through its ZZ[76] and TAZ2 domains[77]. Would then NCP engagement also activate p300? In accordance with this possibility, it has been shown that the TAZ2 domain is required for efficient acetylation of H3K27, while curbing activity towards other lysine residues within nucleosomes[77]. To test this model, we generated homogeneous 'structure-grade' NCPs as shown by single particle cryoEM (Supplementary Fig. 5f, Supplementary Table 4). p300 variants containing the TAZ2 domain, as well as the isolated TAZ2 domain directly bound to NCPs in EMSA experiments (Supplementary Fig. 5c–e). Addition of increasing amounts of NCPs stimulated auto-acetylation of p300s but not p300 core (Fig. 4f).

Cross-linking mass spectrometry experiments showed that surface residues from the p300 Bromo-, RING, PHD and TAZ2 domains directly engage the N-terminal regions of Histone H3 and H2A (Supplementary Fig. 5i, j and Supplementary Data 1). Label free quantitative

mass spectrometry experiments showed that p300 core acetylated most lysine residues indiscriminately in Histones H2B, H3 and H4 while p300s showed a more selective acetylation pattern (Supplementary Fig. 5g, h and Supplementary Data 2). We therefore conclude that direct NCP binding through the TAZ2 domain can stimulate p300 activation but contributes modestly to substrate specificity.

To further test the autoinhibition model, we truncated amino acid residues Δ1813-1838 of the TAZ2 domain (Fig. 4a). Deletion of the C-terminal helix resulted in the formation of p300 nuclear condensates (Fig. 4g) that are typically seen upon p300 activation[47]. p300 mutations found in cancer cell lines that truncate the TAZ2 domain (Supplementary Fig. 5k) also result in p300 activation[78,79]. Together, we conclude that the TAZ2 domain autoinhibits p300 HAT activity. Interaction of TF ADs with the TAZ2 domain allosterically regulates p300 HAT activation by relief of autoinhibition. Mutations that interfere with TAZ2 autoinhibition are found in cancer.

## Mechanism of condensation of BRD4-NUT and p300

BRD4-NUT forms hyperacetylated chromatin megadomains in patient tumours[63,65,68]. Formation of such megadomains can be reproduced by expression of BRD4-NUT in heterologous cell lines[63,65,68]. As there is strong evidence that this system reproduces the phenotype seen in

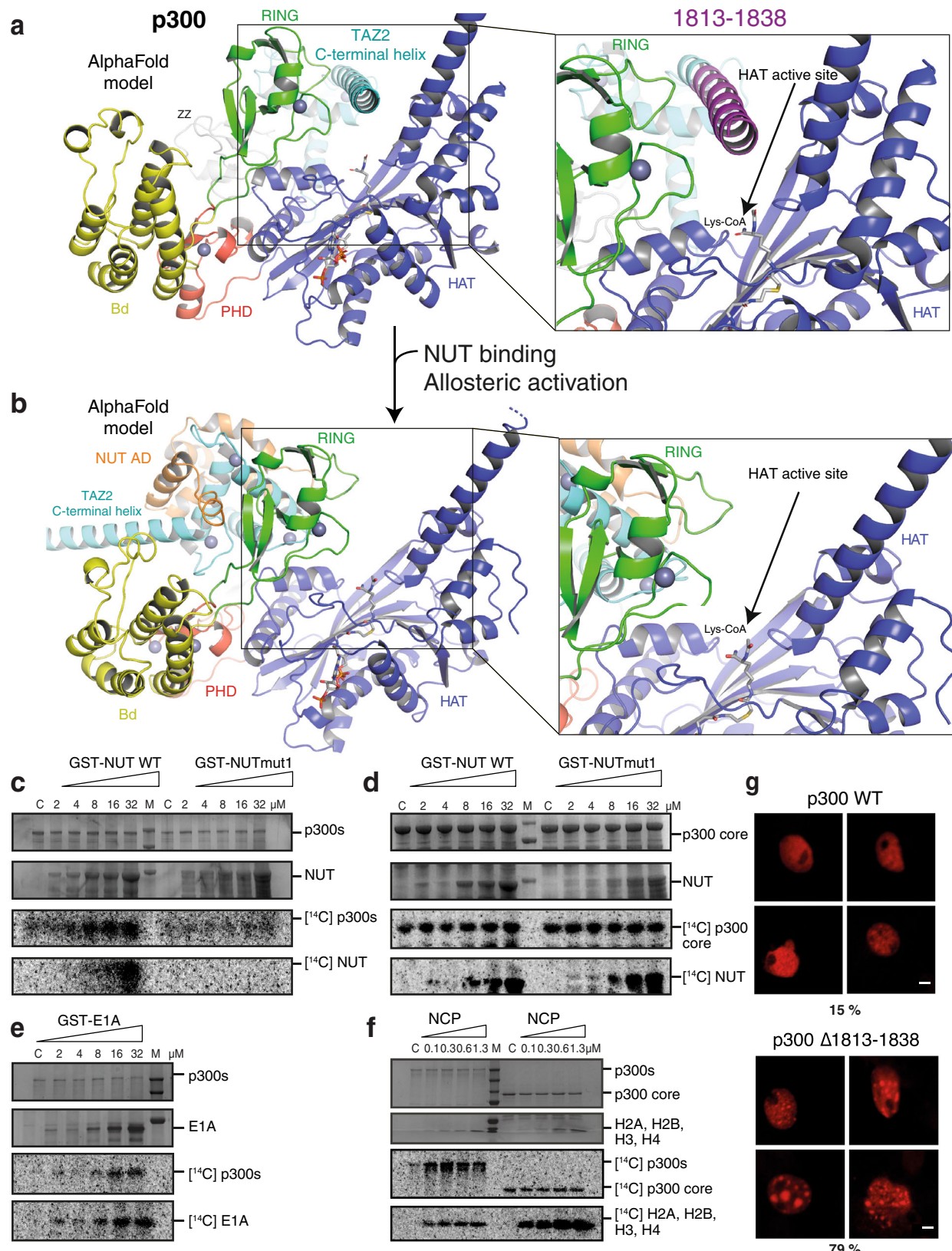

patient cells, we consider it as physiologically relevant, at least with respect to the biophysical properties of the condensation reaction. We generated a series of GFP-BRD4-NUT variants and assessed their propensity to interact with p300 and form co-condensates in the cell nucleus. Co-immunoprecipitation experiments showed that wild-type GFP-BRD4-NUT but not the AD mutants bound to endogenous p300

(Fig. 5a). In agreement with our in vitro data, the wild-type but not mutant variants showed a signal at the level of GFP-BRD4-NUT/p300 when probed with an anti-acetyl lysine antibody (Fig. 5a). GST-pulldowns in nuclear extracts using a purified GST-NUT8wt fragment showed signal for p300 only for the wild-type but not the AD mutant variants (Fig. 5b). Thus, mutation of the NUT AD interferes with p300

**Fig. 4 | Role of the TAZ2 domain in p300 regulation and substrate acetylation.**
**a** AlphaFold prediction of p300 core-CH3 (1048-1838) structure. The structure is coloured according to domain structure. The TAZ2 domain is positioned in the HAT active site thus autoinhibiting substrate access. The TAZ2 C-terminal helix deletion (Δ1813-1838) is coloured in magenta **b** NUT AD binding displaces the TAZ2 domain from the HAT active site thus enabling allosteric HAT activation. **c** p300s or **d**,p300 core were incubated with [$^{14}$C]Acetyl-CoA in the presence or absence (C: p300 alone) of increasing concentrations of wild type or mut1 NUT8 (396-470). **e** Increasing concentrations of GST-E1a we incubated with p300s. **f** Increasing

concentrations of NCPs were incubated with p300s or p300 core. Panels c-f: Top two images: Analysis by SDS–PAGE followed by Coomassie staining. Bottom two images: $^{14}$C phosphor imaging. Experiments c-f were repeated independently three times with consistency. Representative data are shown. Source data are provided as a Source Data file. **g** Cos7 cells were transfected with the HA-tagged p300 WT (top) or p300 Δ1813-1838 (bottom) and analysed by anti HA immunofluorescence. Four representative nuclei are shown. The percentage of cells showing condensates (n = 100 cells) is indicated. Experiments were repeated twice with consistency. Scale bars, 5 μm.

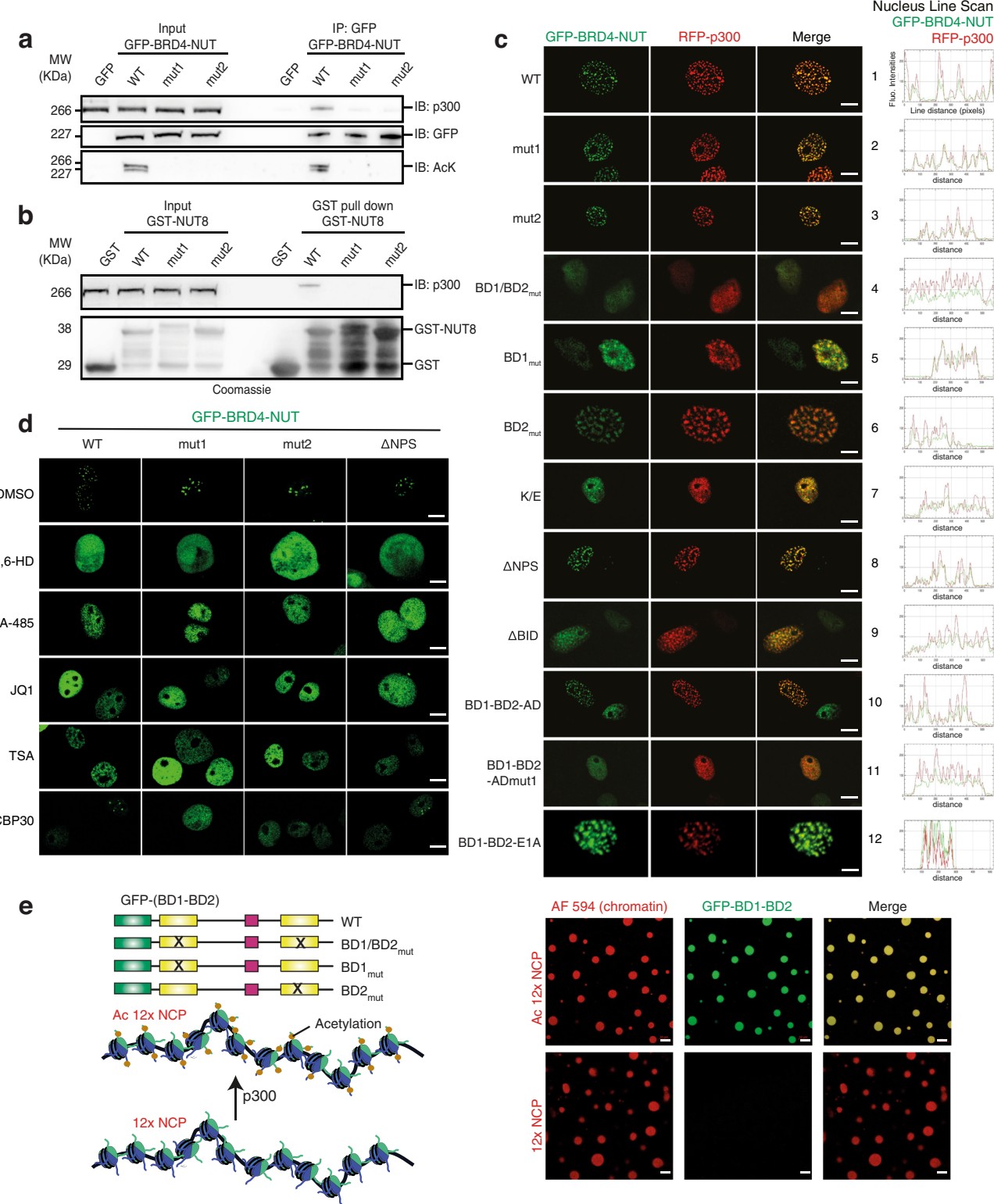

**Fig. 5 | Mechanism of condensation of BRD4-NUT and p300. a** Cos7 cells were transfected with GFP-BRD4-NUT variants or GFP alone as the control. Co-immunoprecipitation (IP) experiments were performed using anti-GFP antibody and immuno-blotted (IB) for p300, GFP and Acetylated-Lysine. Input samples were loaded as a control. **b** Purified GST-NUT8 wild type and mutants 1 and 2 were incubated with Cos7 cells extract for 4 h. The complexes were pull-down by Glutathione Sepharose beads and analysed by SDS-PAGE followed by Coomassie staining (bottom) or by IB for p300 (top). Source data are provided as a Source Data file. Experiments in a-b we repeated independently at least three times. Representative data are shown. **c** Cos7 cells were transfected with the indicated GFP-BRD4-NUT and RFP-p300 constructs and analysed by fluorescence microscopy, BRD4-NUT (green) and p300 (red). Scale bars, 10 μm. Quantification of foci formation of different BRD4-NUT variants and their co-localization with p300 are shown in Supplementary Tables 5, 6. **d** Cos7 cells were transfected with the indicated GFP-BRD4-NUT constructs and treated with DMSO (control), 1,6-hexanediol, p300 HAT inhibitor A485, pan-BET inhibitor JQ1, HDAC inhibitor TSA or p300 bromodomain inhibitor CBP30. Scale bars, 10 μm. Quantification of foci formation in presence of CBP30 is shown in Supplementary Table 6. **e** Schematic representation of GFP-labelled BRD4 mutants and Alexa Fluor 594 labelled reconstituted chromatin containing non-acetylated (12x NCP) or acetylated (Ac 12X NCP) 12 Nucleosome Core Particles. Fluorescence microscopy images of non-acetylated or acetylated 12x NCP chromatin incubated with GFP-BD1-BD2. Experiments in c-e were repeated independently at least three times with consistency. Scale bars, 10 μm.

interaction and BRD4-NUT acetylation in cells, as also seen with the corresponding fragments in vitro (Fig. 4c).

Co-transfection experiments showed that GFP-BRD4-NUTwt and RFP-p300 formed nuclear co-condensates (Fig. 5c; row 1; for quantification see Supplementary Tables 5, 6) both in co-transfected and RFP-p300 negative cells. Thus, condensation of GFP-BRD4-NUTwt not only concentrates but also depletes p300 from other areas of the nucleus. These condensates were independent of the NUT AD as mutant variants still formed condensates that co-localized with p300 (Fig. 5c; row 2&3) and acetylated chromatin (Supplementary Fig. 6a). Mutation of the acetyl-lysine binding pocket in BD1 (N140A) and BD2 (N433A) completely abolished GFP-BRD4-NUT and RFP-p300 co-condensation (Fig. 5c; row 4). To assess the model that BRD4 multivalency drives the condensation reaction[58], we mutated BD1 and BD2 individually. We found that mutation of BD1 but not BD2 largely abolished condensation (Fig. 5c; row 5, 6). Thus, the acetyl-lysine binding ability of BD1 is required to nucleate BRD4-NUT and p300 co-condensation.

We reported previously that BRDT interacts with nucleosomes through its first (BD1), but not second (BD2) bromodomain, and that acetylated Histone recognition by BD1 is complemented by a direct BD1–DNA interaction[80]. Another report indicates that DNA binding promotes condensation of a short BRD4S 'isoform' in vitro[60]. We therefore introduced charge inversion mutations in a conserved, basic surface patch in BD1 (R68E, K72E, K76E) that is involved in DNA binding in BRDT[80] and in the equivalent BD2 amino acids (S358E, K362E, A366E). Mutation of the basic patch abolished condensation (Fig. 5c; row 7). Thus, we propose that BRD4 BD-DNA interaction facilitates acetylation-dependent NCP binding and chromatin condensation in cells.

We also probed the BRD4 N-terminal cluster of phosphorylation sites (NPS) and the basic residue-enriched interaction domain (BID), elements that have been proposed to enable BRD4 regulation by phosphorylation[81]. Deletion of the NPS did not interfere with condensation while removal of the BID reduced condensation (Fig. 5c; row 8, 9). A minimal BD1-BD2 variant fused to the AD but not an AD Mut1 variant (Fig. 1b) still formed p300 co-condensates (Fig. 5c; row 10, 11, Supplementary Movie 1). The BD1-BD2-AD fusion protein but not the AD Mut1 variant was still able to co-IP p300 from cells indicating that the ability to interact with p300 is maintained in the fusion protein (Supplementary Fig. 6c). Fusion of the minimal BD1-BD2 variant to the AD of the Adenoviral E1A (Fig. 1b) also resulted in co-condensation with p300 (Fig. 5c; row 12). We therefore conclude that, at a minimal level, both BRD4 BD acetyl-lysine binding and TF AD-mediated p300 recruitment and activation are required for co-condensation.

Addition of 1,6-hexanediol, the p300/CBP HAT inhibitor A-485 or the BET inhibitor JQ1 all abolished condensation of GFP-BRD4-NUTwt, mut1, mut2 and ΔNPS confirming that both acetylation by p300/CBP and BRD4 BD substrate engagement are required (Fig. 5d, Supplementary Tables 7, 8). HDAC inhibition results in expansion of Histone acetylation and retargets lysine-acetyl readers including BRD4 thus partially mimicking the effect of BD inhibition[82]. In agreement with this model, addition of the HDAC inhibitor Trichostatin A abolished condensation of all variants (Fig. 5d). Immunofluorescence experiments further showed that the BRD4-NUT condensates colocalise with HDAC1 thus confirming that these sites correspond to high acetylation turnover (Supplementary Fig. 6b).

As mutation of the NUT AD did not abolish p300 recruitment into the condensate we further assessed the contribution of the CBP/p300 BD. We used the CBP30 inhibitor that has-34-fold higher selectivity for CBP/p300 BD as compared to BRD4 BD1[83]. Addition of CBP30 did not impact the condensation of GFP-BRD4-NUTwt and ΔNPS. However, condensation of the NUT AD mutants GFP-BRD4-NUTmut1 and mut2 was sensitive to CBP30 treatment (Fig. 5d, Supplementary Table 8). Thus, the NUT AD mutants reveal that co-condensation can also occur through a p300 BD-acetyl-lysine-dependent recruitment mechanism. Thus, two interactions drive BRD4-NUT and p300 co-condensation: (1) interaction of p300 with the AD domain in NUT and (2) p300 BD-acetyl-lysine engagement. Disruption of both interactions, but not each interaction individually, abolishes the co-condensation reaction.

**Condensation requires BD multivalency**

Our observation that mutation of BD1 but not BD2 largely abolished BRD4-NUT condensation in cells (Fig. 5c; row 5, 6) suggests that condensation in cells is not simply driven by BRD4 BD divalency. We therefore tested the requirements of acetylation-dependent chromatin condensation in vitro. Previous data show that addition of BRD4 or an engineered multivalent protein containing five copies (BD1)$_5$ of BRD4 BD1 induces condensation of p300-acetylated nucleosome arrays in vitro[58].

We generated dodecameric nucleosome arrays (Fig. 5e, Supplementary Fig. 6d). In agreement with previous data[58], non-acetylated nucleosome arrays formed round droplets that were dissolved by p300 acetylation (Supplementary Fig. 6e), in agreement with the model that acetylation disrupts internucleosomal interactions[84]. As expected, addition of GFP-(BD1)$_5$ allowed droplet formation of acetylated but not of non-acetylated nucleosome arrays (Supplementary Fig. 6f). This condensation reaction was driven by acetyl-lysine binding as a N140A mutation in the BD1 acetyl-lysine binding pocket abolished droplet formation (Supplementary Fig. 6g).

As a minimal BRD4 construct containing BD1-BD2 (amino acids 1-459) induced condensation in cells when fused to the NUT AD, we tested if BD1-BD2 divalency is sufficient to drive droplet formation in vitro. BRD4 BD1-BD2 enabled droplet formation of acetylated but not of non-acetylated nucleosome arrays (Fig. 5e, right) in the same way as the GFP-(BD1)$_5$. Mutation of the BD acetyl-lysine binding pockets individually or in combination abolished droplet formation (Supplementary Fig. 6g). We conclude that the BD divalency of BRD4 is sufficient to drive condensation in vitro. As large-scale condensation in cells is sensitive to BD1 but not BD2 mutation (Fig. 5c), our model is that BRD4 oligomerisation contributes to condensation.

**Dynamics of condensation of BRD4-NUT and p300**

Time-lapse imaging showed that the condensates remained stable during the experiment (Fig. 6a, Supplementary Movie 2). Addition of

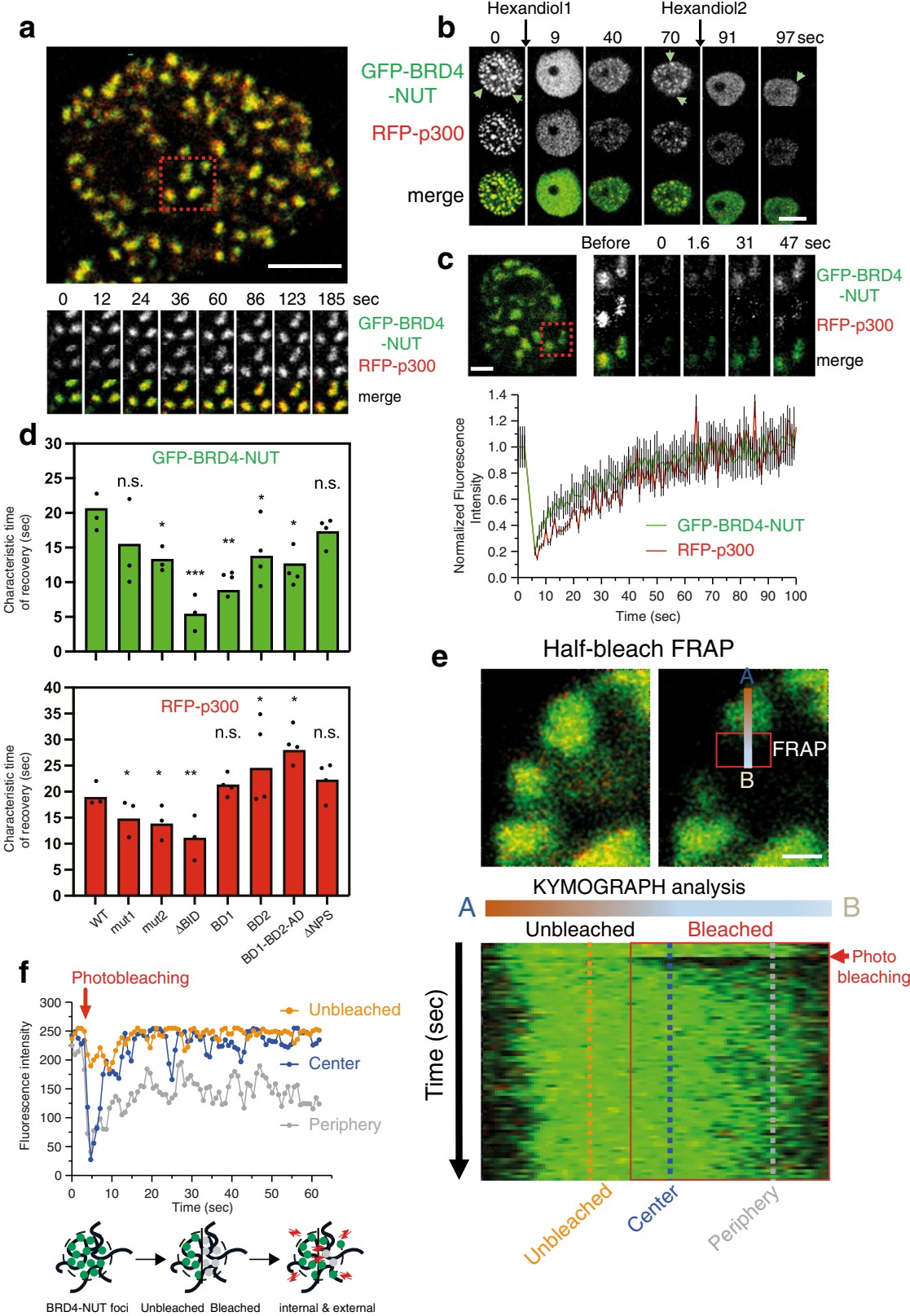

1,6-hexanediol resulted in rapid and transient dissolution of the condensate (Fig. 6b). After reformation, a second addition of 1,6-hexanediol dissolved the condensates again illustrating their dynamic and transient nature. To study the consequences of the BRD4-NUT mutations on turnover of the condensate we performed Fluorescence Recovery After Photobleaching (FRAP) experiments. Wild-type GFP-

BRD4-NUT and RFP-p300 recovered ~50% of signal after ~20 s (Fig. 6c). Thus, while droplets appear stable at the mesoscale, the molecules are highly mobile at the nanoscale and in constant exchange with the nucleoplasm. Mutation of BRD4-NUT mostly interfered with droplet retention as shown by increased FRAP recovery rates (Fig. 6d). p300 dynamics was directly coupled to that of the BRD4-NUT mutants

**Fig. 6 | Dynamics of condensation of BRD4-NUT and p300. a** Example image of nuclear droplets. Red square: Time laps imaging of BRD4-NUT and p300 droplets; Scale bar, 4 μm. **b** Time laps imaging after addition of 3% [w/v] 1,6 Hexanediol. Scale bar, 12 μm. Experiments in a-b were repeated independently two times with consistency. **c** Top: Example image of droplets (red square) after photobleaching. Scale bar, 2 μm. Bottom: Quantification of the FRAP experiments. $N = 3$ independent technical replicates were done. Averages and standard error of mean (SEM; grey bars) are shown. **d** Quantification of the characteristic time of recovery rates for different GFP-BRD4-NUT mutants (green, top) and RFP-p300 (red, bottom). $N = 3$–4 technical replicate FRAP experiments were done for each mutant (40 to 65 foci measured). Averages recovery rates for each N replicate experiment are shown as dots (•). Bar height represents the mean recovery rate value. Statistical differences in recovery rates were determined by a two-sided Student's $t$-test. *$p \leq 0.05$; **$p \leq 0.01$; ***$p \leq 0.001$; n.s. non-significant difference as compared to WT. **e** Half-bleach FRAP experiment. Top: The area before (left) and after (right) laser bleach are shown. The red box indicates the bleached area. Scale bar, 0.5 μm. Bottom: The change in fluorescence intensity along the A-B axis was analysed in the Kymograph. **f** Top: Plot of fluorescence recovery from the Kymograph for the unbleached area (orange), centre (blue) and periphery (grey) of the droplet. Bottom: cartoon of the internal and external exchange. Experiments in **e**, **f** were repeated twice with consistency.

except for the GFP-BRD4-NUT BD2 and the BD1-BD2-AD constructs which showed slower p300 recovery rates indicating some degree of functional uncoupling.

To assay internal droplet dynamics, we performed half-bleach FRAP experiments (Fig. 6e). We analysed the change in fluorescence intensity over time along a defined A-B axis in the bleached (red square) and unbleached area. Kymograph analysis showed that the fluorescence signal in the 'unbleached' region (orange line) and the droplet 'centre' (blue line) recovered much quicker and to completion as compared to the 'periphery' of the droplet (grey line), which only partially recovered signal (Fig. 6f). Our interpretation is that recovery at the centre occurs more quickly because high local concentration in the droplet centre maintains a feed-forward acetylation reaction that traps quickly diffusing molecules. At the droplet periphery, lower concentrations and competition with HDACs destabilizes droplet assembly leading to reduced recovery rates. As there is (1) constant exchange with the nucleoplasm and (2) no preferential equilibration between the bleached and unbleached area we argue that these condensates are 'liquid'-like at the nanoscale but not at the mesoscale where the viscoelastic properties of the 'solid' chromatin scaffold constrain dynamic exchange[85]. As a result, we do not see droplet fission or fusion at least during the time-scale of our experiments.

## Discussion

Chromatin acetylation has long been linked to transcriptional activation, by altering chromatin structure and by enabling binding of BD containing proteins[86–88]. How acetylation promotes transcription has remained elusive. p300 is responsible for up to a third of nuclear acetylation[35]. The BRD4-NUT fusion reveals how constitutive activation of p300 establishes a self-organizing, acetylation-dependent feed-forward reaction that results in large-scale intra- and inter-chromosomal interactions with an oncogenic outcome[62]. In addition to its well-known role in enhancer function and control of gene expression[32,49], p300 is also involved in compartmentalisation of the genome into eu- and heterochromatin[33]. We argue that key aspects of these reactions involve modulation of an acetylation-dependent condensation reaction we describe here.

Regulation of p300 is central for establishment of an acetylation-dependent condensation reaction (Fig. 7a). We propose that in the deacetylated state p300 is auto-inhibited by the RING, TAZ2 domains and the AIL. p300 is activated by trans-autoacetylation[47,89,90]. Due to low p300 concentration[34], self-collision and activation do not occur spontaneously. Instead, as HDACs are estimated to be ~100-fold more abundant[34], p300 most frequently collides with HDAC complexes[91,92] which maintain p300 in the inactive conformation. In this inactive state p300 is evenly dispersed in the nucleus[47].

Induced proximity, by TF binding, enables p300 self-collision, trans-autoacetylation and activation[47]. The BRD4-NUT system reveals an autoinhibitory role for the TAZ2 domain which is relieved upon NUT AD binding (Fig. 4). Accordingly, the well-characterized TAZ2 ligand E1A[93,94] also stimulates p300 auto- and substrate acetylation (Fig. 4), as also reported for other TAZ2 ligands[95,96]. Thus, we propose that the TAZ2 domain plays a role in allosteric HAT regulation: TF AD binding

results in displacement of the TAZ2 domain from its auto-inhibitory position resulting in HAT activation thus directly coupling TF AD-mediated p300 binding to HAT activation.

Perturbation of the autoinhibitory function of the RING and TAZ2 domains by mutation all lead to constitutive p300 activation and condensation reactions similar to what is seen with BRD4-NUT[44,47]. Importantly, comprehensive studies of cancer cell lines show that truncation of the TAZ2 domain (Supplementary Fig. 5k) leads to p300 HAT gain-of-function in disease[78,79]. An acetylation-dependent condensation reaction may also explain how Adenoviral E1A causes genome-wide relocalisation of p300 and activation of genes that promote cell growth and division[97–99]. Not unlike BRD4-NUT, this condensation reaction is also sensitive to p300 HAT, BD and HDAC inhibition[99]. Therefore, while p300 is thought of as a classic tumour suppressor, with loss of function mutations occurring in a number of different cancers[100], constitutive activation by mutation or by interaction with activating ligands such as BRD4-NUT and E1A can lead to acetylation-dependent condensation with an oncogenic outcome.

As numerous TFs interact with p300 through their ADs[31,45], we predict that the condensation reaction establishes intra- and inter-chromosomal interactions that are normally involved in enhancer signalling (Fig. 7b). Many TF ADs, including NUT (Fig. 1), are IDRs that form dynamic interactions with the different protein interaction domains of p300[75,101–110]. Due to such multivalent AD interactions, p300 is able to scaffold TF assembly on complex enhancer elements thereby facilitating combinatorial control of enhancer-mediated transcription[111–113].

TF-mediated recruitment and activation of p300 result in local histone and TF acetylation and establishment of positive feed-back through binding by BD containing proteins. High rates of self-collision of p300 maintain this positive feedback and establish a self-organizing feed-forward reaction resulting in co-condensation of p300, BET proteins BRD2, BRD3, and BRD4 and numerous transcriptional and chromatin regulators[66]. The condensation reaction is limited by HDACs which are usually present at active enhancers and promoters[114,115], and prevent hyperacetylation and transcriptional inhibition[116,117]. HDAC1 is localised in the BRD4-NUT condensate (Supplementary Fig. 6b). Our FRAP experiments show that the condensates are 'liquid'-like at the nanoscale and we propose that HDACs enable continuous and dynamic turnover. If the feed-forward condensation reaction is sufficiently strong, it will overcome this inhibitory effect by HDACs provided that local high concentrations are maintained. As a result, the BRD4-NUT condensate is essentially stable at the mesoscale despite continuous internal dynamic exchange. In enhancer signalling, this dynamic bistability may then render the condensate more immediately responsive to dynamic and transient changes in TF signalling and explain the short transcriptional burst kinetics seen in cells[118,119].

The reaction redistributes p300 (and BD containing proteins) into the condensate thus leading to depletion from other genomic regions (Fig. 5)[63]. The condensation reaction thus leads to high local acetylation (in the condensate) but global histone hypoacetylation elsewhere[120]. Counterintuitively, treatment with HDAC inhibitors does not lead to BRD4-NUT 'hypercondensation'. Instead, HDACi restores

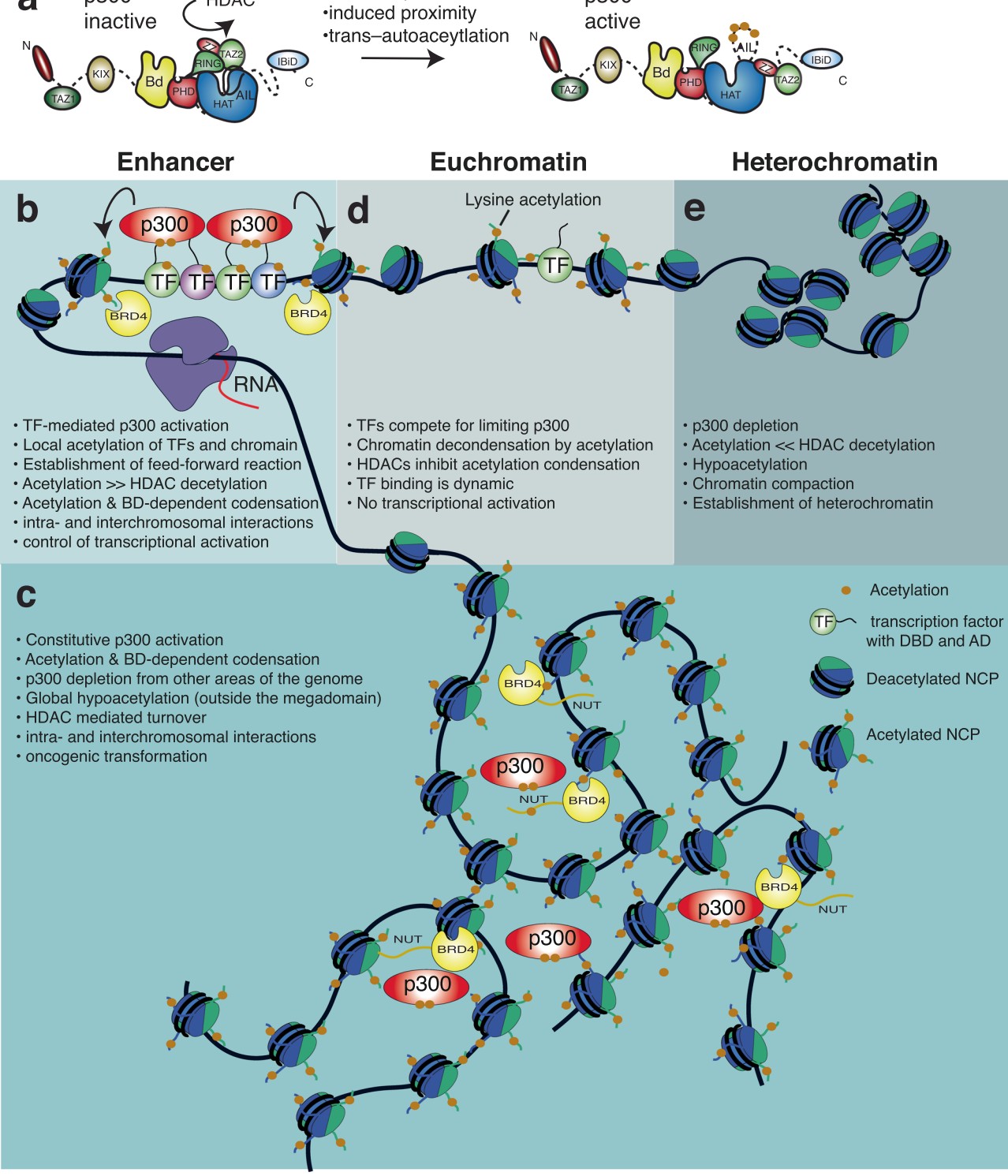

**Fig. 7 | Molecular model for p300 activation in physiological setting and disease. a** p300 is maintained in the inactive state by HDACs. Activation requires TF binding which results autoacetylation and removal of autoinhibition. **b** In enhancer signalling, TFs interact through their ADs with the different protein interaction domains of p300 which results in scaffolding of dynamic TF assemblies. Local TF and histone acetylation allows binding of BD containing proteins including BRD4 thus establishing positive feed-back. HDACs are located in these condensates and enable dynamic turnover thus resulting in bistability and response to changes in TF signalling. **c** In BRD4-NUT NMC, the acetylation-dependent reaction drives formation of intra- and interchromosomal interactions resulting in the formation of a Megadomain. **d** TF binding and acetylation establish open Euchromatin. TFs compete for limiting p300 co-activators but TF-DNA binding outside enhancer clusters remains transient and, due to high HDAC activity, does not effectively establish an acetylation-dependent condensate. **e** Depletion and partitioning of p300 into euchromatic compartments results in hypoacetylation due to high HDAC activity. Hypoacetylated compartments become a substrate for heterochromatinization.

global histone acetylation[120] and rapidly dissolves the BRD4-NUT/p300 condensate (Fig. 5). In agreement with recent data[82], our model is that HDAC inhibition and resultant increased global histone acetylation redistributes BD containing proteins thereby decreasing their effective local concentration resulting in droplet dissolution. Due to the diffusible nature of the reactants, maintenance of global histone hypoacetylation is therefore a key aspect of how such condensates arise. We therefore argue that HDACs contribute to transcriptional regulation by maintaining (1) dynamic bistability of enhancer condensates and (2) global hypoacetylation which in turn 'enables' focal enhancer condensation. Thus, the dual nature of HDAC function with roles in gene activation and repression[114,121–123] would then be directly linked to modulation of these condensation reactions.

While histone H3K27 acetylation has long been seen as a signature mark for active enhancers[124,125], p300 also acetylates numerous non-histone substrates including many TFs[35]. Therefore, we do not see a role for a defined 'code' in how such modifications contribute to the signalling reaction[126]. Instead, promiscuous lysine acetylation of TFs and histones drive emergent multivalency that kinetically traps diffusing BD-domain containing factors such as p300 and BRD4. While individual BDs typically have weak binding affinity and specificity[127], local proximity facilitates rebinding after dissociation thus enabling an acetylation-dependent feed-forward co-condensation reaction (Fig. 6). We propose that this biophysical mechanism enables formation of an 'exploratory' network of interactions that generates bistability by positive feedback and enables focal accumulation of regulatory factors on enhancers and establishment of long-range interactions that control promoter activity[119]. Successful establishment of this 'exploratory' network could then explain the long-standing observation that multiple TF DNA binding sites synergistically activate transcription without the requirement of physical interaction between TFs[112,128].

It has long-been known that TFs activate some genes while at the same time repressing others through competitive coactivator 'squelching'[30,129–135]. Accordingly, in BRD4-NUT, the sequestration of p300 inhibits p53 and the apoptotic cell response while knock-down of BRD4-NUT releases p300, and restores p53-dependent cell differentiation and apoptosis[63]. Thus, we predict that competition for diffusible co-activators and successful establishment of a continual acetylation-dependent condensation reaction is a key regulatory control mechanism: only enhancers containing the appropriate set of clustered TF binding sites will effectively compete and establish such an acetylation-dependent condensate. As TFs locate their target DNA sites using 'facilitated diffusion'[136,137], we propose that following TF-DNA binding, the combined action of p300 scaffolding, TF acetylation and formation of an acetylation-dependent condensate slows down the dynamic search behaviour thus increasing local concentration, residency time and transcription activation. Accordingly, acetylation-mediated increase of TF residence time has been described for p53[138,139] and could thus control TF turnover and transcriptional regulation[140,141].

Binding events that occur outside TF enhancer clusters are then unable to compete for limited co-activators, fail to establish such an acetylation-condensation reaction and therefore remain transcriptionally inactive[142–144]. Due to high concentrations of HDACs[34] and rapid deacetylation kinetics across the genome[35,91], the acetylation-dependent condensation reaction is transient. It is such transient and dynamic acetylation that establishes euchromatic compartments without necessarily leading to transcriptional activation (Fig. 7d).

There is increasing evidence suggesting a role for LLPS of intrinsically disordered proteins in different nuclear processes[39,59,60]. IDR-driven condensation reactions of chromatin-binding proteins including BRD4 and p300 have been implicated in gene regulation[37–39,41,42,49,59,60] and in disease[145,146]. As BRD4-NUT and p300 are largely disordered (Fig. 1; 72% and 42% predicted IDR, respectively) our assumption was that IDR-mediated interactions would contribute to the condensation reaction.

While the NUT AD region can indeed form spherical condensates in vitro as analysed by SAXS (Supplementary Fig. 7), we do not think that this is relevant for the condensation reaction in cells. Even a minimal BRD4 BD1-BD2 construct, removing a large fraction of IDR sequence from BRD4-NUT, enables condensation in cells provided it contains a functional AD (Fig. 5c). As an equivalent NUT AD mutant (that fails to bind to the TAZ2 domain) does not enable the condensation we exclude the possibility that residual structural disorder, e.g. the 200 amino acid linker between the BD1 and BD2 domains of BRD4, drives the condensation reaction. We and others have shown that the condensation reaction is inhibited by (1) the BET BD inhibitor JQ1[63,67], (2) mutation of the acetyl-lysine binding pocket of BRD4 BD1 or, (3) the DNA binding surfaces of BRD4 BD1/BD2 or, (4) p300 HAT inhibition (Fig. 5). In vitro, divalency of BD1-BD2 was essential for condensation of acetylated nucleosome arrays (Fig. 5e). BET proteins are thought to oligomerize[50,56] and BRD4 phosphorylation enables dimerization[147]. We propose that the combination of BD divalency and BET oligomerization provide multivalency that enables condensation of acetylated chromatin in cells. Several p300 HAT gain-of-function mutations, that interfere with the autoinhibitory function of the RING and TAZ2 domains, enable a similar acetylation- and BD-dependent condensation reactions[47]. We therefore conclude that the reaction is driven by p300 activation and engagement of acetylated chromatin by the BDs. In accordance, single-gene imaging data support the notion that promoter-enhancer communication and transcription control depend on TF and BRD4 clustering driven by specific DNA and BD binding rather than IDR-mediated interactions[61].

The co-activator depletion reaction may also be one of the key reaction mechanisms that enables the physical segregation of transcriptionally active euchromatin from repressed heterochromatin (Fig. 7e), and explain the general link between the transcriptional state and 3D chromatin compartmentalisation[148]. Targeting of heterochromatin with acidic TF ADs or treatment with HDAC inhibitors generally counteracts heterochromatinization even in the absence of active transcription[17–23,149–151]. TF-mediated retention of p300 in euchromatin enables establishment of heterochromatin in worms[33], while depletion of p300 and chromatin hypoacetylation facilitates methylation by PRC2[91]. Thus, competition for and partitioning of limiting key acetylases by TFs into euchromatin and depletion from heterochromatin, together with BD-mediated higher order chromatin folding[50,51,56], may drive key aspects of 'epigenetic' genome compartmentalization. Phenomena such as stochastic and exclusive choice of monoallelic gene expression or X-chromosome inactivation in mammals may then also depend on depletion of limiting coactivators and establishment of positive feedback so that only the 'winning' allele drives gene expression while 'competing' alleles are silenced[152–156].

## Methods

### Constructs

cDNA encoding residues for NUT-F1c (347-588), NUT1 (347-446), NUT2 (427-527), NUT3 (508-588), NUT4 (347-480), NUT5 (347-514), NUT6 (351-470), NUT7 (448-492) (wild type and mutants), NUT8 (424-492) (wild type and mutants), NUT9 (395-482) and p53 (1-61) were cloned into the *NcoI/XhoI* restriction sites of vectors pETM-30 or pETM11 (EMBL) with a tobacco etch virus (TEV)-cleavable N-terminal His$_6$ glutathione S-transferase (His$_6$-GST) tag for pETM30 or an N-terminal His$_6$-tag for pETM11. The p300 TAZ2 domain constructs (TAZ2 1723-1812, TAZ2 1723-1822, TAZ2 1723-1836) were codon optimized and cloned into the *NcoI/BamHI* restriction sites of pETM-11 (EMBL) with a TEV-cleavable N-terminal His$_6$-tag. Cysteines which do not participate in zinc coordination were mutated to Alanine or Serine (C1738, C1746, C1769, C1770). The CH3 domain of p300 (1660-1815) was cloned using *EcoRI/XhoI* restriction sites in pGEX-5×1 vector with a TEV-cleavable N-

terminal glutathione S-transferase (GST) tag. p300s (324-2094) and p300 core-CH3 (1043-1828 and 1043-1817) were produced as described previously[44]. The BRD4 constructs: GFP-BD1-BD2, GFP-BD1-BD2 N140A/N433A, GFP-BD1-BD2 N140A and GFP-BD1-BD2 N433A were cloned into the *NcoI/XhoI* restriction sites of pETM11. The vectors expressing the GFP-(Bromo)5 and the GFP-(Bromo)5 N140A were gifts from Michael Rosen (University of Texas Southwestern). The co-expression vector encoding the *S. cerevisiae* histones was a gift from Stephen Harrison (Harvard Medical school). The vector expressing the E1A protein was a gift from Andy Turnell (University of Birmingham). For cell-based assays, all the GFP-tagged BRD4-NUT variants were cloned into pEGFP-C1 vector (Clontech), RFP-tagged NUT-F1c, NUT1, NUT2, NUT3 and NUT4 and RFP tagged p300 and HA-tagged p300 constructs (WT, Δ1813-1838) were cloned into pcDNA3.1. All constructs were confirmed by DNA sequencing.

## Expression and purification

Expression and purification of Flag-tagged p300s and GST-tagged p300 core-CH3 constructs was done as reported previously[44,47]. All p300 TAZ2 constructs were produced as inclusion bodies in *Escherichia coli* (*E. coli*) BL21 (DE3). Cells were grown to mid-log phase (OD600 ~ 0.6) at 37 °C in Luria Broth containing 50 μg/ml kamamycin and 200 μM ZnSO₄. Protein expression was induced by addition of 1 mM isopropyl-β-D-thiogalactopyranoside (IPTG) at 18 °C for 16 h. Uniformly ¹⁵N- and ¹⁵N/¹³C-labelled samples were produced from cells grown in M9 minimal medium containing 50 μg/ml Kanamycin, 200 μM ZnSO₄ and 1 g/l of ¹⁵NH₄Cl and 2 g/l ¹³C glucose. Cells were harvested by centrifugation for 20 mins at 5000 g in a SLC-6000 rotor (Sorvall) and resuspended in 50 ml/L culture lysis buffer 1 (20 mM Tris-HCl pH 7.5, 300 mM NaCl, 5 μM ZnSO₄, 1 mM Dithiothreitol (DTT)), supplemented with EDTA-free protease inhibitors (Roche), and 10 μl of benzonase nuclease (Sigma-Aldrich). Cells were sonicated and centrifuged in a SS-34 rotor (Sorvall) for 1 h at 12,000 rpm. The pellet was then washed three times in the lysis buffer and the final pellet resuspended in buffer 2 (20 mM Tris pH 7.5, 300 mM NaCl, 1 mM Dithiothreitol (DTT), 6 M urea) for 1 h at 4 °C on a rotating device. The protein solution was centrifuged as above and the soluble fraction applied to Ni²⁺-charged IMAC Sepharose 6 FF (Cytiva) beads pre-equilibrated with buffer 2 containing 20 mM Imidazole. The column was washed with 20 column volumes (CVs) of buffer 2 containing 20 mM Imidazole and the bound protein eluted with the buffer 2 containing 300 mM Imidazole. The protein was refolded by dialysis overnight against buffer 3 (20 mM Tris pH 7.0, 100 mM NaCl, 200 μM ZnSO₄ and 2 mM DTT).

The refolded TAZ2 was incubated with His-tagged TEV protease (1:100 w/w) for 14-16 h at 4 °C. Subtractive Ni-NTA chromatography was used to remove the residual His-tag and TEV protease. The cleaved TAZ2 was further purified on a 5 ml HiTrap SP HP cation exchange column (Cytiva) preequilibrated in buffer 4 (20 mM Tris pH 7.0, 100 mM NaCl, 5 μM ZnSO₄ and 1 mM Dithiothreitol (DTT)) and eluted using a 20 CV NaCl gradient. The peak containing TAZ2 was further purified using a HiLoad 16/60 Superdex 75 column (Cytiva) pre-equilibrated with buffer 5 (20 mM Tris pH 7.0, 300 mM NaCl, 5 μM ZnSO₄ and 1 mM DTT). The purified protein was concentrated using a prewashed Amicon Ultra-10 concentrator (molecular weight cut off 10 kDa; EMD Millipore), flash frozen in liquid N2 and stored in −80 °C.

Unlabelled, ¹⁵N- and ¹⁵N/¹³C-labelled NUT fragments were produced as insoluble inclusion bodies in *E. coli* and purified as described above. After Ni-NTA purification, the protein was dialysed and refolded overnight at 4 °C against 1 L buffer 5 (20 mM Tris pH 7.0, 300 mM NaCl, 1 mM DTT and 10% glycerol). Refolded His-GST-NUT fusion proteins were incubated with His-tagged TEV protease (1:100 w/w) for 14-16 h at 4 °C. Subtractive Ni-NTA chromatography was used to remove the residual His-GST-tag and TEV protease. The cleaved NUT proteins were applied to a 5 ml HiTrap Q HP anion exchange column (Cytiva) in

buffer 4 without ZnSO₄ and eluted in a 20 CV NaCl gradient. The peak containing NUT was further purified by gel filtration on a Superdex 75 Increase 10/300 GL column (Cytiva) equilibrated in buffer 5 without ZnSO₄. The final proteins were concentrated in a prewashed Amicon Ultra-15 Centrifugal filter (molecular weight cut off 3 kDa; EMD Millipore), flash-frozen in liquid nitrogen and stored at −80 °C.

The GST-E1A amino acids 1-289 and GST-p53 AD amino acids 1-61 proteins were expressed in *E. coli* BL21 (DE3) cells. Cell pellets were resuspended and sonicated in lysis buffer (20 mM Tris-HCl pH 7.5, 300 mM NaCl, 0.5 mM TCEP), supplemented with EDTA-free protease inhibitors (Roche), and 10 μl of benzonase nuclease (Sigma-Aldrich) and lysed by sonication. The lysate was clarified by centrifugation for 1 h in a SS-34 rotor (Sorvall) at 12,000 rpm and applied to a Glutathione Sepharose 4 Fast Flow resin according to instructions from the manufacturer (Cytiva). The resin was washed with 20 CV lysis buffer and the protein was eluted in lysis buffer containing 20 mM reduced L-Glutathione. The protein solution was then dialysed overnight against lysis buffer to remove L-Glutathione and then applied to a 5 ml HiTrap Q HP anion exchange column (Cytiva) and further purified by gel filtration on a Superdex 75 Increase 10/300 GL column (Cytiva) as previously described for the NUT fragments.

The BRD4 GFP-BD1-BD2, GFP-BD1-BD2 N140A/N433A, GFP-BD1-BD2 N140A and GFP-BD1-BD2 N433A were all expressed in *E. coli* BL21 (DE3) cells. The cells were sonicated in lysis buffer (20 mM Tris-HCl pH 7.5, 300 mM NaCl, 0.5 mM TCEP) and the lysate was clarified as previously described for the other proteins. The clarified lysate was applied to Ni²⁺ charged IMAC Sepharose 6 FF (Cytiva) beads pre-equilibrated with lysis buffer containing 20 mM Imidazole. The column was washed with 20 column volumes (CVs) of lysis buffer and the bound proteins eluted with the lysis buffer containing 300 mM Imidazole. Imidazole was removed by dialysis overnight against 1 L lysis buffer and the proteins were loaded on a 5 ml HiTrap SP HP cation exchange column (Cytiva) as described for TAZ2 and further purified using a HiLoad 16/600 Superdex 200 column (Cytiva) preequilibrated with 20 mM Tris pH 7.0, 300 mM NaCl and 1 mM DTT. The purified proteins were concentrated using a prewashed Amicon Ultra-10 concentrator (molecular weight cut off 30 kDa; EMD Millipore), flash frozen in liquid N2 and stored in −80 °C.

The GFP-(Bromo)5 and GFP-(Bromo)5 N140A were purified as described previously[58]. The co-expressed *S. cerevisiae* histones octamer was purified as described previously[157]. NUT peptides were ordered from Proteogenix.

## HAT assays

The p300 autoacetylation and substrate acetylation assays were performed using [¹⁴C] acetyl-CoA (Perkin-Elmer). Autoacetylation of p300 was quantified by autoradiography after SDS-PAGE gel analysis. The acetylation reaction was performed for 30 min at 30 °C in 1x HAT buffer (20 mM Tris-HCl pH 7.5, 100 mM NaCl, 1 mM DTT, 10 % glycerol and 1x Complete EDTA-free protease inhibitor (Roche)) with 2.5 μM p300s or p300 core and 50 μM [¹⁴C] acetyl-CoA with or without increasing concentrations of the different substrates (NUT fragments, NCP, E1A). 5 μl of the reaction was quenched by addition of 2.5 μl of 3x SDS gel loading buffer followed by analysis on a 4-20% SDS-PAGE gel. The gels were stained with Quick Coomassie Stain (Generon), fixed for 30 min in enhancer solution (KODAK ENLIGHTNING Rapid Autoradiography Enhancer, Perkin Elmer) and dried for 2 h using a Bio-Rad Gel Dryer. The ¹⁴C signal was quantified using phosphor imaging (Typhoon, Cytiva).

## In vitro nucleosome acetyltransferase assays

Nucleosome acetylation reactions were performed in 20 μl volume in acetylation buffer (20 mM Tris pH 7.5, 100 mM NaCl, 1 mM DTT, 10% glycerol and complete Protease Inhibitor EDTA-free (Roche)) with 50 μM Acetyl-CoA, 2.5 μM of p300s or p300 core and 2 μM of

nucleosome core particles (NCPs). Control experiments contained only NCPs and Acetyl-CoA without p300. The reactions were incubated for 30 min at 30 °C and stopped by addition of 1x SDS loading buffer. Each condition was performed in duplicate. The samples were then analyzed by SDS-PAGE and mass spectrometry.

## Multi angle laser light scattering-size exclusion chromatography

SEC-MALLS was performed at a flow rate of 0.5 ml/min on a Superdex 75 Increase 10/300 GL column equilibrated in buffer (20 mM HEPES pH 7.0, 300 mM NaCl, 5 µM ZnSO$_4$, 0.5 mM TCEP). 50 µl of TAZ2, NUT4 or the TAZ2:NUT4 complex at 2 mg/ml were injected onto the column and multi angle laser light scattering was recorded with a laser emitting at 690 nm using a DAWN-EOS detector (Wyatt Technology Corp). The refractive index was measured using a RI2000 detector (Schambeck SFD). The molecular weight was calculated from differential refractive index measurements across the centre of the elution peaks using the Debye model for protein using ASTRA software version 6.0.5.3.

## Biolayer interferometry assay

BLI measurements were performed with a ForteBio Octet RED384 instrument and ForteBio biosensors. Data analyses was done using ForteBio Data Analysis 9.0 software. Kinetics assays were carried out at 25 °C using settings of Standard Kinetics Acquisition rate (5.0 Hz, averaging by 20) at a sample plate shake speed of 1000 rpm. The GST-tagged NUT8 wild type and mutants were loaded onto Anti-GST biosensors and the TAZ2 protein was used as an analyte. The GST-NUT-loaded Anti-GST biosensors were dipped in kinetic buffer (1x PBS, 0.1% BSA, 0.02 % Tween 20) to establish a baseline time course and then dipped into wells containing TAZ2 (1723-1836) at various concentrations diluted in the same kinetic buffer to monitor NUT-TAZ2 binding. The dissociation step was monitored by dipping the biosensors back into the wells used to collect the baseline time course. To monitor binding due to nonspecific interactions of TAZ2 with the biosensors, Anti-GST biosensors were used as controls for every analyte concentration and a double referencing method was used for data subtraction and analysis. The subtracted binding curves were analysed and plotted as a steady-state response to obtain the dissociation constant $K_d$ value.

## Limited proteolysis

To identify stable variants of NUT we performed limited proteolysis by Trypsin and Chymotrypsin. 100 µl of the purified CH3-NUT-F1c complex (250 µg) and TAZ2-NUT4 complex (250 µg) were incubated at 4 °C with Trypsin or Chymotrypsin with a ratio of 1:100 (w/w). At the indicated time points (30 min, 2 h, overnight), 30 µl of the reaction was quenched by addition of 15 µl of 3x SDS gel loading buffer followed by analysis on 4-20 % SDS-PAGE gel. The stable fragments on gel were analysed by acid hydrolysis and mass spectrometry at the Proteomics Core Facility (EMBL, Heidelberg).

## SAXS analysis

X-ray scattering data were collected on the NUT-F1c sample at 100 µM in phase separation buffer: 20 mM Tris pH 7.0, 2 mM DTT, 125 mM NaCl and 10% (w/v) PEG 6000 at the B21 BioSAXS beamline of the Diamond Light Source (Chilton, Oxfordshire, UK). The data were analysed with the ATSAS package (Supplementary Table 9)[158]. Data collected from the sample were substracted from buffer scattering. R$_g$ values were obtained from the Guinier approximation sR$_g$ < 1.3 using Primus[159]. Distance distribution functions p(r) were computed from the entire scattering curve using GNOM[159].

## In vitro NUT phase separation assay

Phase separation of purified GST-NUT-F1c in vitro was done phase separation buffer. The concentrations of GST-NUT-F1c used were

0-200 µM. A GST control was used in the same concentration range under identical conditions. The droplets were visually observed and turbidity of the sample (OD$_{600}$) quantified using NanoDrop (Thermo scientific).

## GST pulldowns

For GST pulldowns, 50 µg of GST-tagged NUT constructs were mixed at an equimolar concentration with binding partners in 50 µl pulldown buffer (20 mM HEPES pH 8.0, 300 mM NaCl, 0.5 mM TCEP, 5 µM ZnSO$_4$, 0.1% Triton X-100) containing 25 µl of a 50 % slurry of GST Sepharose 4 Fast Flow beads (Cytiva) per reaction. Reactions were incubated for 30 min at 4 °C. Twenty-five microliters of the reaction were withdrawn as the reaction input and the remainder was washed 5 times with 500 µl of pulldown buffer. Samples were boiled in 1x sample loading buffer for 5 min and analysed by SDS-PAGE.

## Luciferase reporter assays

Cos7 cells were co-transfected for 36 h with β-Galactosidase reporter plasmid and UAS-promoter luciferase reporter plasmid either alone or with GAL4 DNA binding domain (DBD) plasmid or increasing amount of GAL4-DBD-NUT-F1c plasmid. Cells were then washed and incubated 15 min at RT with Luciferase lysis buffer (1% Triton X-100, 10% Glycerol, 2 mM EDTA, 25 mM Tris-HCl pH 7.8). Luciferase activity was measured directly after mixing 10 µl of cellular extract with 50 µl of Luciferase substrate (Luciferase assay kit, Stratagene). Normalization was made by measuring the β-Gal activity after 1-h incubation of 7.5 µl of cellular extract with 50 µl of β-Gal substrate (β-Gal detection kit II, Clontech).

## Proteomics of histone modifications

For the identification and quantification of histone acetylation following in vitro HAT assays, we followed the previously published procedure[52,53]. In brief, reconstituted NCPs were incubated with 100 µM acetyl-CoA, or in addition purified p300s (340-2,094) or the p300 core domain (1,043-1,666) for 30 min at 30 °C. HPLC-MS/MS was used to quantify site specific H2A, H2B, H3 and H4 acetylation by label-free quantification. Histones were in-gel digested by trypsin (1:20, Promega Corp., Madison, WI, USA) for 16.5 h in 100 mM ammonium bicarbonate (Sigma-Aldrich, Inc., St. Louis, MO, USA) buffer. To make sure loading amounts are consistent among different samples, equal amounts of proteins were subjected to digestion by trypsin. The resulting tryptic peptide were desalted and loaded onto a home-made column packed with 12 cm length × 3 µm ID C18 resin (Dr. Maisch GmbH, Beim Bruckle, Germany). LC-MS/MS was performed on an Orbitrap Exploris™ 480 mass spectrometer (Thermo Fisher Scientific, Inc., Waltham, MA, USA) coupled with an EASY-nLC 1000 system (Thermo Fisher Scientific, Inc., Waltham, MA, USA). Mobile phase A was 0.1% formic acid in water, and mobile phase B was 0.1% formic acid in acetonitrile (v/v). The eluting flow rate was 0.3 µL/min. Samples were separated and eluted with a gradient of 5% to 35% mobile phase B in A over 20 min. Under the positive-ion mode, full-scan mass spectra were acquired over the m/z range from 300 to 1,400 using the Orbitrap mass analyzer with mass resolution of 60,000. MS/MS fragmentation is performed in a data-dependent mode, of which the 15 most intense ions are selected for MS/MS analysis at a resolution of 15,000 using collision mode of HCD. Other important parameters: isolation window, 2.0 m/z units; default charge, 2 +; normalized collision energy, 30%; maximum IT, auto; AGC target, standard; dynamic exclusion, exclude after 2 times within 20 s.

Database search and label-free quantification were performed by Proteome Discoverer 2.5. MS/MS spectra were searched against the reverse-concatenated non-redundant FASTA Saccharomyces

cerevisiae database compiled from the UNIPROT database with incorporation of the reconstituted histone sequences. Six variable modifications (−131.040 Da for N-terminus methionine loss, +15.995 Da for methionine oxidation, +42.011 Da for lysine and N-terminal acetylation, +14.01565 Da for lysine and arginine monomethylation) were used for database search. All other parameters were set as default including 10 ppm and 0.02 Da for the precursor and fragmentation tolerances, respectively. After peptide-spectra matching, 1% FDR was applied to assembly and select high-confidence identifications as default. The peptide intensities were normalized and scaled for label-free quantification. Normalization was performed according to the abundances of all peptides. The peptide-spectrum matches (PSMs) for all acetylated histone peptides were manually verified as described[160]. Two technical replicate experiments were done.

### Isolation and purification of 1 × 601 Widom DNA

Carrier plasmids of 1x Widom 601 repeats were purified using QIAGEN Plasmid Giga kit from 6 L LB culture according to the manufacturer's instructions. 1x arrays DNA were cut from carrier plasmid DNA by incubation overnight with 5,000 units of *EcoR*V-HF restriction endonuclease (New England Biolabs) in 1x CutSmart Buffer (20 mM Tris,OAc, pH 7.9, 50 mM KOAc, 10 mM Mg[OAc]$_2$, 0.1 mg/mL BSA). The excised DNA fragments were separated from the linearized plasmid by PEG precipitation. 0.192 volume of NaCl and 8% PEG 6000 were added to the DNA sample and incubated for 1 h on ice, the sample was then centrifuged for 20 min at 4 °C, 27,000 g. the *EcoR*V fragments contained in the supernatant were precipitated by the addition of 2.5 volumes of 100% cold ethanol. The DNA fragments were air dried for 10 min and dissolved in buffer TE (10 mM Tris/HCl, pH 8.0, 1 mM EDTA). Purity of the extracted 12 × 601 and 1 × 601 fragments were verified on 0.7% agarose gels and the concentration measured by 260 nm absorbance on a NanoDrop spectrophotometer (Thermo Scientific).

### Isolation and purification of 12x Widom 601 array DNA

12×601 DNA template was prepared as described previously[161]. In short, a plasmid containing 12-mer DNA with 177 bp repeats of the 601 sequence separated was expressed in *E.coli* DH5α. The DNA was extracted by alkaline lysis and purified on a Sepharose-6 column. The 12-mer 601 array was separated from the plasmid by *EcoR*V-HF digestion and PEG6000 precipitation.

### Histone octamer preparation

Wild type human histones (H2A, H2B, H3 and H4) and the cysteine mutant Xenopus H4 S47C were individually expressed in *E. coli* strain Rosetta 2 DE3 pLysS and purified following the protocol described[162], but using a RESOURCE S 6 ml cation exchange chromatography column. For octamer reconstitution, equimolar ratios of the four histones were refolded together in the presence of 5 mM β-mercaptoethanol. The octamer was then separated from dimers by gel filtration using a Superdex 200 column (Cytiva). Two batches of octamers have been generated: one fully wild-type human octamer (unlabelled octamer) and an octamer formed by wild-type human H2A, H2B, H3 and Alexa Fluor™ 594 C5 Maleimide labelled *Xenopus* H4 S47C (labelled octamer).

### Electrophoretic mobility shift assays

For EMSA analysis, reconstituted NCPs and indicated concentrations of p300 (core-CH3, core and TAZ2) were maintained on ice for 30 min in EMSA buffer (10 mM HEPES pH 7.5, 150 mM NaCl, 2 mM DTT, 1 mM MgCl$_2$, 200 μM ZnSO$_4$, 0.005% Tween-20), and reactions were then analyzed on 5% acrylamide native gels using 0.25x TBE (Tris, Boris Acid, EDTA) running buffer for 30 min at 4 °C. The reconstituted 12x nucleosome arrays were analyzed on 2% agarose gels in 0.5x TBE running buffer for 2 h at 4 °C. All gels were pre-run prior to samples

loading for 30 min at 4 °C. The gels were stained with SYBR Safe (Thermo Fisher Scientific) to visualize DNA-bound complexes or Coomassie Blue for protein staining.

### Fluorescent labelling of histones

The *Xenopus* H4 S47C mutant histone was labelled using Alexa Fluor™ 594 C5 Maleimide following the protocol from ref. 58. In short, lyophilised H4 S47C was resuspended in 7 M urea, 500 mM Tris-HCl pH 8 and incubated with Tris-neutralized TCEP to a final concentration of 2 mM for 1 hr and 30 min at RT. Alexa Fluor™ 594 C5 Maleimide stock solution was added dropwise to a 1.5 molar excess and the mix was incubated for 4 hr at RT in the dark. The reaction was quenched by the addition of β-mercaptoethanol. The histone was then purified from the free dye by ion exchange chromatography using a HiTrap SP HP cation exchange column followed by dialysis into ultra-pure water with 5 mM β-mercaptoethanol.

### Mononucleosome assembly

DNA and *Saccharomyces cerevisiae* histones octamer were mixed in a 1:0.5 molar ratio. Nucleosomes were reconstituted by salt gradient dialysis method and dialysed against buffer 20 mM Tris-HCl, pH 8.0, 1 mM EDTA, 1 mM DTT with continuous decrease of NaCl concentrations (1.5, 1, 0.6 and 0.25 M) at 4 °C, 3 h each dialysis step with the third dialysis step proceeding overnight using mini dialysis units (10,000 Da MWCO, Pierce Slide-a-lyzer). The reconstituted nucleosomes were quality controlled by electrophoretic mobility shift assays on 5% acrylamide native gels with 0.25x TBE running buffer.

### 12x Nucleosome array assembly, acetylation and phase separation

Chromatin was reconstituted in vitro by assembling the 12-mer DNA with histone octamers by step-wise salt dialysis as described previously[161], but using KCl instead of NaCl. The optimal octamer to DNA ratio was determined empirically. A mix of labelled and unlabelled histone octamer was used. The ratio of labelled to unlabelled histone octamer was also determined empirically. The reconstituted 12-mer nucleosome arrays were quality controlled by *Sca*I restriction enzyme digestion to isolate single nucleosome particles and by electrophoretic mobility shift assays on 2% agarose gel in 0.5x TBE. 12×601 nucleosome arrays at 2 μM mono-nucleosome concentration were incubated with 1 μM of the p300 core domain in Acetylation buffer (20 mM Tris-HCl pH 7.0, 100 mM NaCl, 10% glycerol, 1 mM DTT) for 30 min at 30 °C. The non-acetylated chromatin reactions were incubated under identical conditions but without p300 core enzyme. After acetylation, 5 μM BRD4 GFP-Bromo$_5$, GFP-Bromo$_5$ N140A, 10 μM GFP-BD1-BD2, 10 μM GFP-BD1-BD2 N140A, 10 μM GFP-BD1-BD2 N433A or 10 μM GFP-BD1-BD2 N140A/N433A were added to nucleosomal arrays for a final concentration of 1 μM mono-nucleosome in phase separation buffer (25 mM Tris, pH7.5, 150 mM KOAc, 1 mM MgOAc, 5% glycerol, 5 mM DTT, 0.1 mM EDTA, 0.1 mg/mL BSA)[58].

### Microscopy of chromatin droplets

For confocal microscopy image acquisition, a Zeiss LSM 980 Airyscan 2 microscope equipped with a Plan-Apochromat 63x/1.40 Oil DIC f/ ELYRA and ZEN Blue 3 acquisition and analysis software was used. Samples were imaged in 384 well glass bottom plates for microscopy which were prepared as described in ref. 58. In short, microscopy plates were washed with 5% Hellmanex, etched with 1 M NaOH and treated with 25 mg/mL 5 K mPEG-silane. Prior to use, individual wells were passivated by incubation with freshly prepared 100 mg/mL BSA.

### Cross-linking mass spectrometry

Purified solutions of both NCPs and p300 core-CH3 (1043-1828) were diluted to a concentration between 0.5 and 1 mg/ml in 20 mM HEPES

pH 7.0, 300 mM NaCl, 1 mM DTT and 5 μM ZnCl$_2$ and cross linked using a homobifunctional, isotopically-coded crosslinker DiSuccinimidyl Suberate DSS-H12/D12 (Creative Molecules Inc.) at a concentration of 2 mM for 30 min at 35 °C 600 rpm. The cross-linking reaction was then quenched with 0.1 volumes of 1 M Ammonium bicarbonate for 10 min at 35 °C 600 rpm. The sample was then treated with 0.8 volumes of 10 M urea and 250 mM Ammonium bicarbonate as well as 0.05 volumes RapiGest SF Surfactant (Waters, cat. No. 186008090) and sonicated for 1 min in an ultrasound bath. DTT was later added with a final concentration of 10 mM, incubated for 30 min at 37 °C and freshly prepared Iodoacetamide was added to a final concentration of 15 mM and incubated for 30 min at room temperature in the dark. The protein was sequentially digested with Endoproteinase Lys-C and Trypsin. The enzymes were added at an enzyme to substrate ratio of 1:100 for LysC and 1:50 for Trypsin and the reaction incubated for 3-4 h at 37 °C. The reaction was started by LysC followed by Trypsin addition after 4 h. After digestion, the sample was acidified with Trifluoroacetic acid (1% v/v final concentration) at 37 °C for 30 min, spun down at 17,000 g for 5 min and the peptides were fractionated by peptide size-exclusion chromatography. The mass spectrometry experiment and cross-link identification were performed by the EMBL Proteomics core facility.

## Immunoblotting

Western Blot analyses were carried out according to standard procedures using precast Nu-Page 4-12% Gels (NP0323, ThermoFisher) in MES or MOPS buffer. Wet-Transfers were performed in Tris-Glycine Buffer (EU0550, Euromedex) on 0.45 μM nitrocellulose membrane (106-000-16, Amersham Protran) in a Mini-Protean system (1658005EDU, Bio-Rad) maintained cold by ice. After saturation of the membrane in 1X PBS (#14200-067, Gibco), 0.2% Tween 20 (P7549, Sigma) and 5% milk (Regilait) or Bovine albumin Serum (BSA, #04-100-812-C, Avantor) during 1 h at RT, membranes were incubated at 4 °C overnight with an anti-p300 antibody (1:1000, Santa-Cruz, sc-585), anti-GFP (1:5000, Covance, MMS-118R), anti-H4 acetylated (1:1000, Millipore, #06-598), anti-Lysine-acetylated (1:1000, Cell Signalling, #9441), anti-HDAC1 (1:1000, Santa-Cruz #sc-7872), anti-HA (1:4000, Abcam, ab9110). Membranes were washed 3 times in 1X PBS with 0.2% Tween 20 during 5 min and then incubated with a Goat anti Rabbit IgG(H + L)-HRP (Bio-rad, 1:5000), or a Goat anti-Mouse IgG (H + L)-HRP (Bio-rad, 1:10000) for 1 h at RT. Membranes were then washed three times, as for primary antibodies. Signal detection was performed with a Fusion FX Chemiluminescence system (Vilber).

## Immunofluorescence and drug treatment

Cos7 cells were seeded in glass coverslips (Lab-Tek, #1545256, Brand), and GFP and RFP bearing constructs were transfected by Lipofectamine 2000 (11668-013, Invitrogen). Twenty-four hours later, cells were washed with PBS (#14200-067, Gibco) and fixed with 4% paraformaldehyde (HT5012, Sigma) for 15 min, followed by permeabilization with 0.2% Triton-X100 (T9284, Sigma) and 1 × PBS for 10 min at room temperature. After being blocked with filtered 2.5% BSA (#04-100-812-C, Avantor), 0.2% Tween 20 (P7549, Sigma Aldrich), and 1 × PBS solution at RT for 30 min, slides were incubated with antibodies anti-acH4 (1:100, Millipore #06-598), anti-HA (1:200, H6908, Sigma), anti-p300 (1:100, sc-585, Santa-Cruz) or anti-HDAC1 (1:100, Santa-Cruz #sc-7872) in 2.5% BSA, 0.1% Tween 20, and 1 × PBS in a humidified chamber at 4 °C overnight and then washed three times for 5 min each in the antibody dilution buffer. The secondary antibodies were diluted at 1:500 (Alexa Fluor 546 (A11035) or 647 (A21245)) in the same buffer and incubated in a humidified chamber for 45 min at RT, and then washed as for the primary antibodies. The DNA was counterstained by DAPI (1:500), and the slides were mounted in Dako fluorescent mounting medium (S3023, Dako). For the drug treatment assays, 24 h after being transfected with GFP and RFP bearing constructs, Cos7 cells were treated with DMSO (Sigma), 1,6-hexanediol (10%, 1 h, Sigma),

A485 (3 μM, 1 h, Selleck), JQ1 (1 μM, 1 h, Selleck), TSA (0.3 μM, 12 h, Selleck) or CBP30 (1 μM, 1 h, Selleck), then the cells were washed with PBS and fixed with 4% paraformaldehyde for 15 min.

## Co-immunoprecipitation

Cell pellets were lysed in 300 mM NaCl buffer (20 mM HEPES-KOH pH 7.9, 300 mM NaCl, 3 mM MgCl$_2$, 0.1% NP-40, 1X protease inhibitor cocktail (Roche), 1X phosphatase inhibitor cocktail-1 (Sigma) and 10 mM sodium butyrate) for 30 min at 4 °C with rotation. After a 15 min centrifugation at 12,000 g at 4 °C, the supernatants were collected and separated into input and immunoprecipitate (IP) lysates. IP lysates were mixed with GFP-selector (pre-washed with 300 mM NaCl buffer three times, NanoTag) and rotated overnight at 4 °C. After washing with 300 mM NaCl buffer, IPs were released by adding Laemmli sample buffer to beads and boiling for 5 min at 100 °C.

## Fluorescence recovery after photobleaching (FRAP)

Live imaging and fluorescence recovery after photobleaching experiments (FRAP) were performed with the LSM710 Confocor3 confocal microscope (Carl Zeiss), equipped with the Plan-Apochromat C 63X /1.4 oil-immersion objective. FRAP area, time and time interval (1 image every 783 ms) were maintained identical between the different BRD4-NUT mutants and repeat measurements ($N$ = 3–4, 45–65 foci measured per conditions) in order to photobleach 3-6 foci at the same time. To compare recovery kinetics against BRD4-NUT wild-type, FRAP measurements were fitted to a single exponential curve, $I(t) = I_{(0)} + k_1 e^{-k2t}$ (performed with ZEN software; Carl Zeiss Microscopy) to determine the characteristic time of recovery. Values of t (1/ k2) were compiled and statistical significance was tested using the Student's $t$-test (Excel).

## NMR experiments and AlphaFold calculations

NMR data were collected using Topspin 3 Bruker. Backbone resonances of both proteins free in solution were assigned by measuring HNCO, HNCA, and HN(CO)CA spectra on $^{15}$N,$^{13}$C-labelled samples at 600 (NUT7) and 700 (TAZ2) MHz. We were able to assign 87% (95%) of the NH pairs, 72% (82%) of the CO resonances and 88% (96%) of the CA resonances in the backbone of NUT7 (TAZ2).

Side-chain resonances of both proteins free in solution were assigned by measuring HBHA(CO)NH, H(CCO)NH, and CC(CO)NH spectra on $^{15}$N,$^{13}$C-labelled samples at 600 (NUT7) and 700 (TAZ2) MHz. We were able to assign 70% (55%) of the HA resonances, 57% (45%) of the HB resonances, and 45% (53%) of the CB resonances in NUT7 (TAZ2), as well as part of the γ-protons.

To determine the binding site of NUT7 on TAZ2, we measured $^1$H-$^{15}$N HSQC spectra on $^{15}$N-labelled TAZ2 before and after adding unlabelled NUT7. For each of the assigned peaks in the spectra we calculate the chemical shift perturbations (CSP) defined as

$$CSP = \sqrt{\frac{1}{2}\left(\triangle\delta_H^2 + \left(0.14 \cdot \triangle\delta_N^2\right)^2\right)}, \quad (1)$$

in which $\triangle\delta_H^2$ and $\triangle\delta_N^2$ are the chemical shift differences upon adding NUT7 in the $^1$H and $^{15}$N dimensions, respectively. We used the procedure described by Schumann et al.[163] to determine whether measured CSPs are significant or not. We used the same approach to map the binding site on NUT7 monitoring a titration of a solution of $^{15}$N-labelled NUT7 with unlabelled TAZ2 using $^1$H-$^{15}$N HSQC spectra.

Both $^{13}$C- and $^{15}$N-HSQC-NOESY spectra on a sample of $^{15}$N,$^{13}$C-labelled TAZ2 to which one equivalent of unlabelled NUT7 was added and $^{15}$N-HSQC-NOESY spectra on $^{15}$N-labelled NUT7 in admixture with one equivalent of unlabelled TAZ2 were measured using standard pulse sequences. All NOESY spectra were measured at 950 MHz with a mixing time of 100 ms. The AlphaFold computational models shown in Fig. 3e and Fig. 4a were obtained from alphafold.ebi.ac.uk. Models

shown in Fig. 3f and Fig. 4b were calculated using AlphaFold v2.1.1 with multimer model v1 weights[73].

## HADDOCK docking

We used the HADDOCK protocol[70] with experimental information from titrations and NOESY spectra to derive a structural model of the interaction. We used the NMRbox platform[164] to carry out the calculations. In a first step, we used the results of the titration experiments to derive ambiguous distance restraints (AIRs) describing the binding interface, and we used these AIRs as input for HADDOCK to dock extended conformations of NUT7 on the TAZ2 structure derived from PDB 2mzd. The best 200 structures resulting from the calculations were clustered based on the fraction of common contacts (FCC). The resulting clusters were sorted according to their agreement with the AIRs. Structures in the best cluster display the interaction of residues P448-E454 in NUT7 with TAZ2. Thus, we used the Flexible-Meccano algorithm[165] to generate an ensemble of 100 conformers of NUT7, in which the conformation of residues P448-E454 is the same as in the structure resulted from the HADDOCK calculation, whereas the rest of the chain behaves like a statistical coil. Then, we used HADDOCK again to dock this ensemble of structures on the same TAZ2 conformation as above. In this run, in addition to the same AIRs as in the preliminary run, we used distance restraints derived from NOESY spectra between S1726 in TAZ2 and residues L463 and A464 in NUT7. We also used TALOS-N[166] to derive restraints on the backbone dihedral angles of residues P448-E454 from chemical shift values. We repeated this calculation ten times with different initial seeds and velocities. Finally, the best 200 structures from each of the ten runs were clustered based on the FCC. The resulting clusters were sorted according to their agreement with the AIRs and NOE-based restraints. Then, from the best-ranked cluster in each of the runs, we selected the structure with the best HADDOCK score (which is a linear combination of various energy terms and buried surface area), thereby obtaining a final set of 10 structures. Structure visualisation was done in Coot and validation with Phenix and Molprobity. Renderings were done with Pymol or Chimera.

## Cryo-electron microscopy grid preparation, data collection and image processing

Quantifoil R1.2/1.3 Au 300 grids were glow discharged for 5 min at 40 mA on a Quorum Gloqube glow-discharge unit prior to application of the sample. 3 μl of 1 μM final concentration of the mononucleosome particles in buffer 20 mM Tris-HCl, pH 8.0, 250 mM NaCl, 1 mM EDTA, 1 mM DTT were applied to the grid, blotted for 3 s at blot force 10 and plunge frozen into liquid ethane using a Vitrobot Mark IV (FEI Thermo Fisher), which was set at 4 °C and 100% humidity. Cryo-EM data was collected on a Thermo Fisher Scientific Titan Krios G3 transmission electron microscope at the LISCB at the University of Leicester. Electron micrographs were recorded using a K3 direct electron detector (Gatan Inc.) at a dose rate of 15 e⁻/pix/s and a calibrated pixel size 1.086 Å. Focusing was performed over a range between −2.3 and −1.1 μm in 0.3 μm intervals. Image processing of the NCPs was performed as follows: movie stacks were corrected for beam-induced motion and then integrated using MotionCor2[167]. All frames were retained and a patch alignment of 4 × 4 was used. Contrast transfer function (CTF) parameters for each micrograph were estimated by Gctf[168]. Integrated movies were inspected with Relion-3.1 for further image processing (243 movies). Initial particle sets were picked manually, extracted, and subjected to 2D classification in Relion 3.1[169]. Resulting classes corresponding to protein complexes were subsequently employed as references for auto-picking in Relion 3.1. Particle extraction was carried out from micrographs using a box size of 128 pixels (pixel size: 1.08 Å/ pixel). An initial dataset of 80764 particles was cleaned by several iterations of 2D classification, classes exhibiting clear secondary structure features were selected for 3D classification. After iterative

rounds of 3D classifications, well defined classes were selected and global refinement was performed.

## Reporting summary

Further information on research design is available in the Nature Portfolio Reporting Summary linked to this article.

## Data availability

All data and materials generated during this investigation are available upon reasonable request to the corresponding author. Chemical shifts for NUT are available from the BMRB under accession ID 51611. The mass spectrometry proteomics data have been deposited to the ProteomeXchange Consortium via the PRIDE partner repository with the dataset identifier PXD038798. Source data are provided with this paper.

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

## Acknowledgements

We thank Christos Savva and TJ Ragan for CryoEM support. We thank
Nathan Cowieson from the B21 BioSAXS beamline of the Diamond Light
Source; Mandy Rettel for mass spectroscopy analysis; Luke Bailey, Sarah
Northall and Dipti Vashi for help with histone preparations. Michael K.
Rosen for plasmids encoding GFP-(Bomo)5 and GFP-(Bromo)5 N140A;
Stephen M. Hinshaw and Stephen C. Harrison for plasmids encoding *S.
cerevisiae* histones; Andrew Flaus for *H. sapiens* histones; Timothy J.
Richmond for histone H4 mutant plasmids; Andy Turnell for E1A plas-
mids. Work in the S.K. laboratory was supported by ANR Episperm4
programme, by Plan Cancer Pitcher, by MSD Avenir ERICAN pro-
grammes, by the Cancer ITMO (Multi-Organisation Thematic Institute) of
the French Alliance for Life Sciences and Health (AVIESAN) MIC pro-
gramme, by the "Université Grenoble Alpes" ANR-15-IDEX-02 SYMER
and LIFE programmes and by Fondation ARC programme
RF20190208471. This work used the platforms of the Grenoble Instruct-
ERIC center (ISBG; UAR 3518 CNRS-CEA-UGA-EMBL) within the Grenoble
Partnership for Structural Biology (PSB), supported by FRISBI (ANR-10-
INBS-0005-02) and GRAL, financed within the University Grenoble
Alpes graduate school (Ecoles Universitaires de Recherche) CBH-EUR-
GS (ANR-17-EURE-0003). Work in the T.S. laboratory was supported by
the BBSRC (BB/S019510/1; BB/R016275/1). Work in the D.P. laboratory
was supported by grants from the Worldwide Cancer Research [16-
0280], the Wellcome Trust [221881/Z/20/Z], Instruct-ERIC [PID: 18571]
and the Medical Research Council [MR/W001667/1].

## Author contributions

Z.I. designed and performed most experiments, analysed and validated
the data with support by L.F. T.W., O.D., N.H., C.C. and N.R performed
cell biology experiments. N.S. performed NMR experiments. A.R. per-
formed chromatin analyses in vitro. Y.Z. and J.G. performed mass
spectroscopy analysis. T.S., M.B., and S.K. provided supervision, funding
acquisition and commented on the draft. D.P. was involved in con-
ceptualization, supervision, project administration, funding acquisition.
Z.I. and D.P. wrote the original and revised drafts with input from all
authors.

## Competing interests

Y.Z. is a founder, board member, advisor to, and inventor on patents
licensed to PTM Biolabs Inc (Hangzhou, China and Chicago, IL) and
Maponos Therapeutics Inc. (Chicago, IL). The other authors declare no
competing interests.

## Additional information

**Supplementary information** The online version contains
supplementary material available at

Daniel Panne.

**Peer review information** *Nature Communications* thanks Tatiana Kuta-
teladze, and the other, anonymous, reviewer(s) for their contribution to
the peer review of this work. Peer reviewer reports are available.

