## [Peer Review File · Nature Communications]

Editorial Note: This manuscript has been previously reviewed at another journal that is not operating a transparent peer review scheme. This document only contains reviewer comments and rebuttal letters for versions considered at Nature Communications

REVIEWER COMMENTS

Reviewer #1 (Remarks to the Author):

In light of these changes, I think this manuscript should be published. I believe that it will be a good contribution to the field.

Reviewer #2 (Remarks to the Author):

During the revision, authors have adequately addressed this reviewer's concerns. In particular, authors' change in the text of statement/discussion, which is to stress that the AlphaFold based modeling requires additional experimental test, is laudable; also, please mention in text that the mutant IDR is somewhat less stable compared to WT, and inclusion of raw gel images will ease the concern.

With the minor changes above, I would then recommend the publication at Nature comm.

Reviewer #3 (Remarks to the Author):

1. I appreciate the authors responses however my major concern remains unanswered. It is impossible to unambiguously assign the amide resonances of NUT, shown in figure 2a, right black-brown spectra. If the authors were able to do so, this should be documented in a supplementary figure showing strips of triple resonance experiments with ~87% of amides in NUT, as stated by authors, being connected.

2. The $^1\text{H}, ^{15}\text{N}$ HSQC spectra of completely annotated NUT should appear in the Supplementary material. The assignment should be deposited in the BMRB database with two different entries, apo and bound states of NUT.

3. The authors responded: "Here again, we do not understand the implications of the phrase 'The data shown as histogram of chemical shift changes for the NUT peptide is uncertain.' The interaction interface is clearly identified by the largest chemical shifts in both proteins." Again, if the resonances of NUT in figure 2a right are assigned incorrectly, the histogram for NUT, shown in figure 2c, right panel, will obviously show changes for incorrectly assigned residues.

4. The authors responded: "Contrary to the comment above, this analysis relies entirely on experimental data measured from the complex in solution." The problem is that the authors overstate the importance of modeling using HADDOCK and alpha fold algorithms. These are helpful simulations, still they are simulations. I appreciate that the authors include experimental restrains, NOEs, in the modeling, however intermolecular NOEs should be documented in a supplementary figure showing strips of the filtered NOESY experiments.

Point-by-point response:

Reviewer #1: no further comments are requested

Reviewer #2

Please mention in text that the mutant IDR is somewhat less stable compared to WT, and inclusion of raw gel images will ease the concern.

We now state Line 243: ‘NUT mut1 was a bit more instable as compared to the equivalent WT fragment after purification (see Source Data, Fig. 4c).’

Uncropped gels and phosphorimaging results are now shown as Source Data for Fig. 4c.

Reviewer #3:

1. I appreciate the authors responses however my major concern remains unanswered. It is impossible to unambiguously assign the amide resonances of NUT, shown in figure 2a, right black-brown spectra. If the authors were able to do so, this should be documented in a supplementary figure showing strips of triple resonance experiments with ~87% of amides in NUT, as stated by authors, being connected.

It is indeed possible to unambiguously assign the backbone resonances of NUT, using standard triple resonance approaches (see response 2 for examples of strips from the relevant 3D correlation spectra). This short peptide is not the most challenging IDP for such assignment, examples have been published that are well over 400 amino acids in length and standard approaches are in general successful for IDPs of less the 200 amino acids in length.

2. The $1H,15N$ HSQC spectra of completely annotated NUT should appear in the Supplementary material. The assignment should be deposited in the BMRB database with two different entries, apo and bound states of NUT.

All assigned backbone chemical shifts ($1H$, $15N$, $13C$) have now been uploaded to the BMRB. The deposition ID is 51611 - this code has been added to the manuscript (Line 1490 et seq). As an aid to the reader we now show the annotated $15N-1H$ HSQC spectra for NUT in Supplementary Figure 2h, as requested.

3. The authors responded: “Here again, we do not understand the implications of the phrase ‘The data shown as histogram of chemical shift changes for the NUT peptide is uncertain.’ The interaction interface is clearly identified by the largest chemical shifts in both proteins.” Again, if the resonances of NUT in figure 2a right are assigned incorrectly, the histogram for NUT, shown in figure 2c, right panel, will obviously show changes for incorrectly assigned residues.

It is not clear to us why the reviewer assumes that NUT resonances are assigned incorrectly. As stated above, we have uploaded chemical shifts to the BMRB. Strips from the assignment spectra are shown in Supplementary Figure 2 demonstrating the spectral quality which is clearly sufficient for backbone assignment.

4. The authors responded: “Contrary to the comment above, this analysis relies entirely on experimental data measured from the complex in solution.” The problem is that the authors overstate the importance of modeling using HADDOCK and alpha fold algorithms. These are helpful simulations, still they are simulations. I appreciate that the authors include experimental restrains, NOEs, in the modeling, however intermolecular NOEs should be documented in a supplementary figure showing strips of the filtered NOESY experiments.

As stated on Line 211 et seq: ‘The NOESY spectra contain very few cross-peaks that can be unambiguously assigned’. We therefore have not included a table showing NOE values.

Instead, we have added the values directly in the text (Line 215). We have added the NOESY spectra showing the intermolecular cross peaks in Supplementary Figure 3i.

REVIEWER COMMENTS

Reviewer #3 (Remarks to the Author):

Dear Daniel,

I indeed very much appreciate your responses regarding my comments, I do. However, these responses below do not satisfactorily address my concerns regarding unambiguous assignments of NMR resonances in the spectrum shown in Fig. 2a (NUT). The biological and biochemical data you show in this manuscript are exciting but as an NMR spectroscopist, I cannot support one particular aspect- the NUT NMR assignment data without clear confirmation.

The HSQC spectrum of NUT shows a drastic overlap of resonances, so if you think that the NUT assignments figure and the histogram showing changes in chemical shifts per NUT residue are important here, this needs to be well documented. It's not currently.

Obviously, I support your work but not sure how to deal with this NUT NMR data. I also understand the difficulties of working with unstructured proteins (my lab faces this problem all the time too), but that is the reality.

I would be happy to discuss with you or your colleagues, NMR spectroscopists, how to address this issue if that would be appropriate/needed and allowed by the journal and the Editor.

I understand you need intermolecular NOEs for HADDOCK; but you probably don't need the whole assignment of NUT – only definite, certain, unambiguous assignment of the residues in the binding interface. But then you have to remove overall NUT assignments and the NUT histogram panels.

I am commenting on your responses below as well.

With kind regards,

Tatiana Kutateladze

Reviewer #3:

1. I appreciate the authors responses however my major concern remains unanswered. It is impossible to unambiguously assign the amide resonances of NUT, shown in figure 2a, right black-brown spectra. If the authors were able to do so, this should be documented in a supplementary figure showing strips of triple resonance experiments with ~87% of amides in NUT, as stated by authors, being connected.

It is indeed possible to unambiguously assign the backbone resonances of NUT, using standard triple resonance approaches (see response 2 for examples of strips from the relevant 3D correlation spectra). This short peptide is not the most challenging IDP for such assignment, examples have been published that are well over 400 amino acids in length and standard approaches are in general successful for IDPs of less the 200 amino acids in length.

- Daniel, it's irrelevant that other IDR/IDPs have been assigned, it doesn't matter what size, what does matter- the extent of the overlap of crosspeaks. In the case of your protein, NUT, the overlap is substantial, even it's only 45 aa in length.

- I was asking (still ask) to show strip plots of 87% assigned residues in NUT (either as a Suppl Fig or Source data). If you have assigned 87% of 45 NUT resonances, it means 39 strip plots (as in Suppl. Fig. 3f). These plots should be already available.
- Thank you for including Suppl. Fig. 3f that shows strip plots for four residues- are they for TAZ2 or NUT and from which experiment? This figure is too small, please replace with a larger one so labels are readable.

2. The ¹H,¹⁵N HSQC spectra of completely annotated NUT should appear in the Supplementary material. The assignment should be deposited in the BMRB database with two different entries, apo and bound states of NUT.

All assigned backbone chemical shifts (¹H, ¹⁵N, ¹³C) have now been uploaded to the BMRB. The deposition ID is 51611 - this code has been added to the manuscript (Line 1490 et seq). As an aid to the reader we now show the annotated ¹⁵N-¹H HSQC spectra for NUT in Supplementary Figure 2h, as requested.

- Thank you for including Fig 2h but it's unreadable. Please replace with larger figure, so labels can be read.
- Please deposit the bound state of NUT in BMRB

3. The authors responded: "Here again, we do not understand the implications of the phrase 'The data shown as histogram of chemical shift changes for the NUT peptide is uncertain.' The interaction interface is clearly identified by the largest chemical shifts in both proteins." Again, if the resonances of NUT in figure 2a right are assigned incorrectly, the histogram for NUT, shown in figure 2c, right panel, will obviously show changes for incorrectly assigned residues.

It is not clear to us why the reviewer assumes that NUT resonances are assigned incorrectly. As stated above, we have uploaded chemical shifts to the BMRB. Strips from the assignment spectra are shown in Supplementary Figure 2 demonstrating the spectral quality which is clearly sufficient for backbone assignment.

- I question assignments of NUT because currently I don't see clear documentation that supports the assignments. The strip plots of 39 assigned residues are need here.

4. The authors responded: "Contrary to the comment above, this analysis relies entirely on experimental data measured from the complex in solution." The problem is that the authors overstate the importance of modeling using HADDOCK and alpha fold algorithms. These are helpful simulations, still they are simulations. I appreciate that the authors include experimental restrains, NOEs, in the modeling, however intermolecular NOEs should be documented in a supplementary figure showing strips of the filtered NOESY experiments. As stated on Line 211 et seq: 'The NOESY spectra contain very few cross-peaks that can be unambiguously assigned'. We therefore have not included a table showing NOE values. Instead, we have added the values directly in the text (Line 215). We have added the NOESY spectra showing the intermolecular cross peaks in Supplementary Figure 3i.

- Thank you for including one intermolecular NOEs strip plot in Suppl Fig. 3i. This panel is also small and unreadable. Please replace with a larger image so labels can be read.
- Is this the only residue that shows intermolecular NOEs? Please expand legend for this Fig to clarify which protein was used to collect NOESY.

Dear Tatjana,

We now have drafted a point-by-point response to your comments.

Reviewer #3:

1. I appreciate the authors responses however my major concern remains unanswered. It is impossible to unambiguously assign the amide resonances of NUT, shown in figure 2a, right black-brown spectra. If the authors were able to do so, this should be documented in a supplementary figure showing strips of triple resonance experiments with ~87% of amides in NUT, as stated by authors, being connected.

It is indeed possible to unambiguously assign the backbone resonances of NUT, using standard triple resonance approaches (see response 2 for examples of strips from the relevant 3D correlation spectra). This short peptide is not the most challenging IDP for such assignment, examples have been published that are well over 400 amino acids in length and standard approaches are in general successful for IDPs of less the 200 amino acids in length.

Daniel, it's irrelevant that other IDR/IDPs have been assigned, it doesn't matter what size, what does matter- the extent of the overlap of crosspeaks. In the case of your protein, NUT, the overlap is substantial, even it's only 45 aa in length. - I was asking (still ask) to show strip plots of 87% assigned residues in NUT (either as a Suppl Fig or Source data). If you have assigned 87% of 45 NUT resonances, it means 39 strip plots (as in Suppl. Fig. 3f). These plots should be already available.

All strip plots from the HNCA and HN(CO)CA 3D experiments on the NUT7 peptide for the assigned residues are now shown in Supplementary figure 3 (all strips and one zoom showing 4 residues). As the referee can see there is considerable coherence transfer in both experiments, giving rise to both inter- and intra-residue cross-peaks. This is not surprising. This is a small peptide which is eminently assignable using three-dimensional triple resonance experiments that have been standard for a few decades now.

This particular peptide is not challenging by modern standards, where much longer IDPs – comprising hundreds of amino acids, and considerable resonance overlap - are routinely assigned using the same standard approaches. Indeed, as long as magnetization can be transferred during the coherence transfer delays - during which connectivities between and within residues are established - the strategy is really very straight forward and, in many cases, automatable. The most important obstacle to efficient coherence transfer is transverse spin relaxation, which in the case of IDPs is almost negligible due to their high level of intrinsic dynamics. This efficiency is demonstrated very clearly in the figure requested by the referee. We trust that the referee will now be satisfied that assignment has indeed been achieved.

The data are available at the following link if the referee would like to repeat the assignment – <https://filesender.renater.fr/?s=download&token=fd145753-b58e-47dc-9573-aad02f5b2f82>

Thank you for including Suppl. Fig. 3f that shows strip plots for four residues- are they for TAZ2 or NUT and from which experiment?

This figure has now been replaced by the figures described above, as requested by the referee.

This figure is too small, please replace with a larger one so labels are readable. 2. The ¹H,¹⁵N HSQC spectra of completely annotated NUT should appear in the Supplementary material. The assignment should be deposited in the BMRB database with two different entries, apo and bound states of NUT.

We have increased the figure & font size wherever possible. 2. We make the $^1\text{H}, ^{15}\text{N}$ HSQC spectra of annotated NUT available in the Supplement. As mentioned previously, assigned backbone chemical shifts (^1H , ^{15}N , ^{13}C) have been uploaded to the BMRB. The deposition ID is 51611. The assignment of the 1:1 chemical shifts, determined by titration, is now included in its entirety in supplementary Table S2.

3. The authors responded: "Here again, we do not understand the implications of the phrase 'The data shown as histogram of chemical shift changes for the NUT peptide is uncertain.' The interaction interface is clearly identified by the largest chemical shifts in both proteins." Again, if the resonances of NUT in figure 2a right are assigned incorrectly, the histogram for NUT, shown in figure 2c, right panel, will obviously show changes for incorrectly assigned residues.

It is not clear to us why the reviewer assumes that NUT resonances are assigned incorrectly. As stated above, we have uploaded chemical shifts to the BMRB. Strips from the assignment spectra are shown in Supplementary Figure 2 demonstrating the spectral quality which is clearly sufficient for backbone assignment.

I question assignments of NUT because currently I don't see clear documentation that supports the assignments. The strip plots of 39 assigned residues are need here.

As the assignment has now been demonstrated, we assume that the referee now considers the chemical shift titration that is based on this assignment to be valid.

4. The authors responded: "Contrary to the comment above, this analysis relies entirely on experimental data measured from the complex in solution." The problem is that the authors overstate the importance of modeling using HADDOCK and alpha fold algorithms. These are helpful simulations, still they are simulations. I appreciate that the authors include experimental restrains, NOEs, in the modeling, however intermolecular NOEs should be documented in a supplementary figure showing strips of the filtered NOESY experiments.

As stated on Line 211 et seq: 'The NOESY spectra contain very few cross-peaks that can be unambiguously assigned'. We therefore have not included a table showing NOE values. Instead, we have added the values directly in the text (Line 215). We have added the NOESY spectra showing the intermolecular cross peaks in Supplementary Figure 3i.

Thank you for including one intermolecular NOEs strip plot in Suppl Fig. 3i. This panel is also small and unreadable. Please replace with a larger image so labels can be read. - Is this the only residue that shows intermolecular NOEs? Please expand legend for this Fig to clarify which protein was used to collect NOESY.

This Figure is now shown in **Supplementary Fig. 4f**. We have increased Figure & Font size. The NOE shown in the figure was measured using the ^{15}N edited NOESY using $^{15}\text{N}, ^{13}\text{C}$ labelled TAZ2 and unlabelled NUT7. Figure legend has been clarified.

REVIEWERS' COMMENTS

Reviewer #3 (Remarks to the Author):

The authors have now adequately addressed my comments.